**Data Availability Statement:** All relevant data are within the paper and its S1, S2 Figs and S1–S8 Tables.

# Early centralized isolation strategy for all confirmed cases of COVID-19 remains a core intervention to disrupt the pandemic spreading significantly

Nguyen Hai Nam[1,2,3°], Phan Thi My Tien[3,4°], Le Van Truong[3,5°], Toka Aziz El-Ramly[3,6], Pham Gia Anh[3,4], Nguyen Thi Hien[3,7], El Marabea Mahmoud[3,8], Mennatullah Mohamed Eltaras[3,9], Sarah Abd Elaziz Khader[3,10], Mohammed Salah Desokey[3,11], Ramy Magdy Gayed[3,12], Shamael Thabit Mohammed Alhady[3,13], Bao-Tran Do Le[3,14], Do Phuc Nhu Nguyen[3,15], Ranjit Tiwari[3,16], Mohammed Eldoadoa[3,17], Britney Howard[3,18], Tran Thanh Trung[3], Nguyen Tien Huy[19]*

1 Graduate School of Medicine, Kyoto University, Kyoto, Japan, 2 Harvard Medical School, Global Clinical Scholars Research Training Program, Boston, Massachusetts, United States of America, 3 Online Research Club, Nagasaki, Japan, 4 University of Medicine and Pharmacy at Ho Chi Minh City, Ho Chi Minh City, Vietnam, 5 Traditional Medicine Hospital of Ministry of Public Security, Hanoi, Vietnam, 6 Faculty of Medicine, University of Alexandria, Alexandria, Egypt, 7 Emergency Department, Hue City hospital, Hue City, Vietnam, 8 Faculty of Medicine, Tanta University, Tanta, Egypt, 9 Faculty of Medicine for Girls, Al-Azhar University, Cairo, Egypt, 10 Faculty of Medicine, Ain Shams University, Cairo, Egypt, 11 Faculty of Medicine, Aswan University, Aswan, Egypt, 12 Faculty of Medicine, Cairo university, Cairo, Egypt, 13 Faculty of Medicine, University of Gezira, Wad Medani, Sudan, 14 University of California, Los Angeles, Los Angeles, California, United States of america, 15 Epidemiology Department, Institute of Public Health Ho Chi Minh City, Ho Chi Minh City, Vietnam, 16 Faculty of Medicine, Institute of Medicine, Tribhuvan University, Kathmandu, Nepal, 17 Milton Keynes University Hospital, Milton Keynes, United Kingdom, 18 American University of the Caribbean School of Medicine, Cupecoy, Sint Maarten, 19 School of Tropical Medicine and Global Health, Nagasaki University, Nagasaki, Japan

☯ These authors contributed equally to this work.
* tienhuy@nagasaki-u.ac.jp

## Abstract

### Background

In response to the spread of the coronavirus disease 2019 (COVID-19), plenty of control measures were proposed. To assess the impact of current control measures on the number of new case indices 14 countries with the highest confirmed cases, highest mortality rate, and having a close relationship with the outbreak's origin; were selected and analyzed.

### Methods

In the study, we analyzed the impact of five control measures, including centralized isolation of all confirmed cases, closure of schools, closure of public areas, closure of cities, and closure of borders of the 14 targeted countries according to their timing; by comparing its absolute effect average, its absolute effect cumulative, and its relative effect average.

### Results

Our analysis determined that early centralized isolation of all confirmed cases was represented as a core intervention in significantly disrupting the pandemic's spread. This strategy

**Funding:** The authors received no specific funding for this work.

**Competing interests:** The authors have declared that no competing interests exist.

helped in successfully controlling the early stage of the outbreak when the total number of cases were under 100, without the requirement of the closure of cities and public areas, which would impose a negative impact on the society and its economy. However, when the number of cases increased with the apparition of new clusters, coordination between centralized isolation and non-pharmaceutical interventions facilitated control of the crisis efficiently.

## Conclusion

Early centralized isolation of all confirmed cases should be implemented at the time of the first detected infectious case.

## Introduction

In December 2019, the world turned its gaze towards Wuhan, Hubei province, China, due to the emergence of a novel cluster with a pneumonia-like disease [1]. Following exposure histories of reported cases, these patients mostly worked or had contact with traders at the Huanan Seafood Market, where many wildlife species were sold and slaughtered. In a short time, this spreading infection was no longer limited to China but rapidly penetrated other countries such as Vietnam, Japan, and Thailand, which had recorded cases outside of China [2]. Coronavirus disease 2019 (COVID-19), a newly emerging infection raised from the virus mentioned above, is an acute respiratory infectious disease, caused by a new member of the Coronavirus's family, called Severe Acute Respiratory Syndrome Coronavirus-2 (SARS-CoV-2) [3]. On March 11, 2020, the WHO declared a global pandemic due to COVID-19, with 118,319 confirmed cases [4]. Just one month later, the world witnessed more than 1.6 million confirmed cases, with approximately 100,000 deaths worldwide [5]. Since the last two large-scale outbreaks of the two coronavirus-type epidemics in the 21$^{st}$ century, including the Severe Acute Respiratory Syndrome (SARS) outbreak, which was first identified in the Guangdong province of China in 2002 [6]; and the Middle East Respiratory Syndrome (MERS) outbreak, which was first reported in Saudi Arabia in 2012 [7]; this new crisis has once again deeply impacted people around the world. It confused healthcare workers alike, particularly epidemiologists. Despite valuable experiences in dealing with the previous pandemics, controlling an unknown, highlycontagious respiratory disease, caused by a new coronavirus, is still extremely challenging.

In response to the spread of COVID-19, a plethora of control measures had been proposed and recommended by public health authorities and international organizations from countries around the world. A list of non-pharmaceutical interventions at different levels and various strategies had been established to prevent and mitigate the continual spread of COVID-19, including but not limited to social distancing and barriers with the closure of schools and avoiding modes of public transportation, isolation of suspected and confirmed cases, and rapid closure of borders. However, these lockdown policies had multiple negatives. Miles et al. determined that the lockdown cost was 40% higher than the highest benefits from avoiding the worst mortality case scenario at full life expectancy tariff [8]. For mental health, Adams-Prassl et al. recognized that citizens living in states with implementation of lockdowns scored 0.085 standard deviations lower on the standardized WHO-5 mental health index compared to those living in states without a lockdown strategy [9]. Despite its negative impacts, potentially measuring lockdown implementation also effectively proved its ability to aid in controlling the

pandemic, and it remains a question that we hope to elucidate. Thus, our study made an effort to summarize and highlight distinctive features of significant control measures among 14 particular countries, including China, Hong Kong, Taiwan, Singapore, Korea, Japan, the United States, France, Germany, the United Kingdom, Canada, Italy, Spain, and Sweden. With this approach, we hope to contribute a comprehensive understanding of COVID-19 and determine the core interventions which may help to contain this enemy.

## Materials and methods

### Study population

In order to assess the impact of current control measures on the number of new cases during 70 days from 22 January 2020 (when John Hopkins Coronavirus Resource Center began to collect the data of COVID-19) to the end of March 2020, we have collected data from countries that had the highest number of daily confirmed cases. The United States, Canada, and China, with the highest population of general population, were chosen due to their representative characteristics. 5 Asian countries and 5 European countries with the percentage of days having the highest number of daily confirmed cases higher than 60% were chosen for later analysis. Additionally, Sweden, a North European country that decided to live in peace with the virus, was chosen as a European control country, as it chose to go without both lockdown and centralized isolation. Details of the percentage of days having the highest number of daily confirmed cases of 13 countries were outlined in the S1 Table.

### Data collection

Available information from peer-reviewed and respected sources with real-time updated datasets, including the John Hopkins Coronavirus Resource Center (JHU) and WHO Situation Reports, were retrieved and subsequently analyzed. The total number of confirmed cases, deaths, and day-by-day new cases, as well as the first confirmed COVID-19 case and the first community-acquired case within each country, were determined. Regarding Sweden, since the testing policy primarily focused on citizens with symptoms associated with COVID-19 infection and requirements for inpatient hospital care and/or elderly care, people with mild symptoms were not required to contact their healthcare providers [10]. Thus the number of new confirmed cases in Sweden reported from JHU might be lower than the actual new confirmed cases. To address this limitation, we have estimated the number of new confirmed cases in Sweden by using the number of new confirmed cases of neighboring European countries such as Spain, Italy, the United Kingdom, France, and Germany at the moment; without application of any control measures. Our analysis revealed that the actual new confirmed cases in Sweden were 8.47 times higher than reported by JHU (S2 Table).

Afterward, our team parsed out the real-time enrollment data extracted from official governmental announcements, health ministry updates, creditable and verified articles; to point out the control measures issued by each country with respect to the size of its outbreak according to the timeline. We used data collected from January 20, 2020 to May 11, 2020, at which point most countries started to reopen schools and ceased social distancing protocols. Furthermore, we made a systematic summary of all of the available control measures from websites and country-specific sources by categorizing them into four groups: preventive measures, social distancing-related measures, associated government measures, and associated research measures. Lockdown strategies according to each level (1 to 4), were divided into five subsets: [a](centralized isolation of all confirmed cases), [b](closure of schools), [c](closure of public areas), [d](closure of cities), and [e](closure of borders) (S3 and S4 Tables). In China, Taiwan, and Hong Kong, all suspected and confirmed cases were isolated in healthcare facilities and could be

ruled out after a negative SARS-CoV-2 test with at least 24-hour intervals regardless of the number of days of isolation. In South Korea and Singapore, 14 days of mandatory isolation was applied for all suspected and confirmed cases of COVID-19. In Japan, only suspected and confirmed cases with related symptoms were isolated at the hospitals, whereas the remaining cases without any symptoms were strictly quarantined at medical camps or hotels under medical staff supervision during a required time. Even though the policy of centralized isolation of all confirmed cases was modified across these six countries, we assumed these control measures were comparable when applied. Thus, centralized isolation of all confirmed cases is defined as a quarantine of all confirmed cases in hospitals, medical camps, or hotels under the supervision of medical staff and during a time period set forth by each country's government. Closure of schools was applied with the suspension of all cultural, educational, sports, and teaching activities. Closure of public areas consisted of non-essential business, including stores, shopping centers, and services, such as bars, restaurants, cinemas, theaters, as well as the prohibition of mass gathering. Additionally, citizens were requested to remain home and leave only for essential activities, work from home if possible, and maintain 6 feet of social distancing at all times. Closure of cities involved the restrictions on travel with the suspension of all forms of transportation across cities. Closure of borders is defined as the official announcement of the governments regarding the prohibition of entry and exit of all individuals from all entries, including lands, seas, rails, and air routes. Particularly, to address the heterogeneity of the date of implementation of control measures in countries with various states and cities such as the United States, Canada, and China, we assumed that the first date of applying the lockdown strategies is the representative date for these countries. Details of the date of the control measures mentioned above in each city and state of the United States, Canada, and China with reliable references; were summarized in S5 Table.

## Data management

We divided our members into subgroups that were responsible for data collection of assigned countries. Each country was optimally assigned to a member with different nationalities to avoid selection and information biases. Three independent reviewers did data extraction to ensure accuracy and validation. Any discrepancies were dissolved by discussion among members and a senior author to reach a final consensus. The cleaned data was uploaded and stored to a Google drive encrypted by a password. Only the senior author and the supervisor have the right to access all data.

## Data analysis

In the study, we analyzed the impact of each control measure adopted by the governments of the 14 targeted countries, according to each country's timing, to highlight and assess its significance in keeping the outbreak under control. Since vaccines and target drugs were not available in March 2020, current interventions could only control the number of new cases. However, the number of newly infected cases was greatly affected by the number of existing cases due to the human-to-human spread of COVID-19. Consequently, an analysis of the infection trend through the trend of a new case index was applied. Indeed, if an intervention effectively helps control the incidence of new cases, then the new case index of infection will decrease.

In this study, we used the Causal Impact package to analyze the effectiveness of the intervention. It was designed to find the law of cause and effect of the intervention if it is impossible to generate a randomized controlled trial. We used the Bayesian inference method with Markov chains Monte Carlo (MCMC), 20,000 bootstrap times to create control data with the trend

of the original data series before the time of intervention. After the time of data, intervention continued to be created, but with two different trends: (1) from actual data through Bayes reasoning with MCMC and 20,000 bootstrap times created the data series with intervention; and (2) the control data series is used for trending and the probability of pre-intervention data to create a post-period (without intervention assumption).

Then, the two chains formed were compared with each other regarding the absolute effect average, the absolute effect cumulative, the relative effect average, the posterior probability of a causal effect, and the posterior tail-area probability p. The absolute effect average was defined as the average percent decrease rate of the trend after intervention/day. The absolute effect cumulative was determined as the sum of the absolute effect average. The relative effect average was the actual percentage that decreased after the intervention, and the posterior probability of a causal effect is the repeatability of causality. Five control measures, including centralized isolation of all confirmed cases, closure of schools, closure of public areas, closure of cities, and closure of borders, were evaluated among countries. From the literature, the average incubation period for SARS-Cov-2 is 5.2 days with the 95% confidence interval of the distribution at 13 days [11]. However, we encountered a delay in the time reporting of data compared to the testing time in reality. There were some possible reasons for this. Firstly, there were 13 days of average incubation period as mentioned above. Secondly, the differences in time zones between 14 countries could lead to the differences in reporting the data to the public between JHU and these countries. Besides, from April 2020, the JHU database only updated data at the constant time from 3:30–4:00 pm (UTC). Thirdly, there was a lack of simultaneousness in reporting confirmed cases, recovered cases, and death cases of COVID-19. To solve this issue, we conducted the study with the hypothesis that the time between testing and reporting in official statistics could be fluctuating from one day to four days with the details were described in S6 Table. After the analysis, the results showed that the data changes due to the fluctuation of time between testing and reporting in official statistics led to insignificant differences of 0–3.5% (CI 95%). Thus, we decided to select day 13th as a time point for analysis.

To consistently assess across countries, we used the time when the country had a confirmed case of a 10th infection as the first day, except for China. This moment is considered an alert time since many countries initiated adjusting their control measures. Additionally, it facilitated the evaluation of the effectiveness of the centralized isolation of all confirmed cases. China alone used the first day on January 20, 2020, when China was in the central wake of the disease announcement. Therefore, our analysis process adhered to the following assumptions: (1) assess the intervention's effectiveness starting on the 13th day after the intervention was implemented; and (2) countries applying the interventions could simultaneously offer solutions, sometimes very far apart. Thus, we only separated the interventions when the interval between two consecutive interventions was greater than 13 days; and (3) if the interventions were less than 13 days apart, we would only analyze the overall effectiveness of the interventions calculated after 13 days of the last implementation.

## Results

The dates of implementation of the four levels of lock-down strategies at the moment of the 10th confirmed cases are outlined in Table 1. Based on our findings, Korea's data was analyzed at three different moments, which were on the 27th day with centralized isolation of all confirmed cases, the 32nd day with the closure of schools, and the 22nd day with the closure of public areas. With the data for China, we analyzed the 14th day, when the country applied centralized isolation of all confirmed cases, and the 54th day when they closed the entire country. Regarding Japan's data, we analyzed the 16th day of centralized isolation of all confirmed

**Table 1. Date of implementation of the national intervention at the moment of the 10th confirmed cases.**

| Country | Centralized isolation all confirmed cases | | Closure of schools | | Closure of public areas | | Closure of cities | | Closure of borders | |
|---|---|---|---|---|---|---|---|---|---|---|
| | Date | Days | Date | Days | Date | Days | Date | Days | Date | Days |
| **Spain** | NA | | 11 March | 15 | 14 March | 18 | 14 March | 18 | 16 March | 20 |
| **Italy** | NA | | 4 March | 13 | 10 March | 19 | 8 March | 17 | 10 March | 19 |
| **UK** | NA | | 18 March | 24 | 20 March | 26 | NA | | NA | |
| **Canada** | NA | | 14 March | 20 | 15 March | 21 | NA | | 18 March | 24 |
| **US** | NA | | 12 March | 39 | 15 March | 42 | 19 March | 46 | 21 March | 48 |
| **France** | NA | | 16 March | 38 | 14 March | 36 | 22 March | 44 | 17 March | 39 |
| **Germany** | NA | | 16 March | 44 | 16 March | 44 | NA | | 16 March | 44 |
| **China** | 2 February | 14 | 27 January | 1 | 23 January | 4 | 23 January | 4 | 28-Mar | 69 |
| **Korea** | 26 February | 27 | 2 March | 32 | 21 February | 22 | NA | | NA | |
| **Japan** | 14 February | 16 | 2 March | 33 | 16 April | 78 | 16 April | 78 | 8 March | 39 |
| **Singapore** | 23 January | -8 | 8 April | 70 | 27 March | 58 | NA | | 23 March | 54 |
| **Hong Kong** | 26 January | -4 | 25 January | 6 | 28 March | 60 | NA | | 25 March | 57 |
| **Taiwan** | 21 January | -11 | 2 February | 12 | 25 March | 55 | NA | | 18 March | 48 |
| **Sweden** | NA | | NA | | NA | | NA | | NA | |

NA: Not application.

cases of COVID-19, together with the analysis of the 33[rd] day when Japan closed its schools. We evaluated Singapore's data on the 58[th] day when they closed their public areas and the 70[th] day when they closed their schools. Exceptionally, we analyzed the 12[th] day as Taiwan initiated opening their schools and the 55[th] day when they closed their public areas. Hong Kong, Singapore, and Taiwan were not analyzed for the effect of centralized isolations of all confirmed cases because the pre-intervention time was too short. Sweden was excluded from all of the above control measures since this country denied to against the virus. The date of implementation of the national intervention at the moment of the 10th confirmed cases is outlined in Table 1. And its reference link is presented in S7 Table.

Table 2 showed that when the above countries applied neither methods [a,b,c,d,e], pre-intervention new case indices [100% x (new cases/active cases)] ranged from 12.60% to 25.98%. At this time, countries only applied centralized isolation[a] of all confirmed cases without methods [b,c,d,e], including Taiwan (11 days before the 10[th] case), Singapore (8 days before the 10[th] case), and Hong Kong (4 days before the 10[th] case); their pre-intervention new case indices were 8.36%, 8.93%, and 11.38%, respectively. On the other hand, with centralized isolation of all confirmed cases[a], the closure of schools[b] and public areas[c], with or without the closure of cities;[d] showed the effectiveness by decreasing the new case indices in Korea and China after 13 days. When China solely applied method[a], its pre-intervention new case index was 17.29% (12.90%, 22.04%). China's post-intervention index decreased to 0.72% (0.52%, 0.96%) when the country applied methods[a,b,c,d]. South Korea's pre-intervention new case index was 13.98% (9.13%, 19.93%), and its post-intervention index decreased to 1.34% (1.12%, 1.58%) with the induction of methods[a,b,c]. An illustration of the change in the new case index according to the application of different intervention strategies was represented in S1 Fig.

Applying measures of social distancing simultaneously, including [b](closure of schools), [c](closure of public places), and [e](closure of borders) without [d](closure of cities); helped Canada, France, and Germany to reduce their absolute effect averages (AEA) to10% (p<0.001), 15% (p<0.01), and 11% (p<0.01), respectively. Notably, the AEA of Spain, Italy, and the United States reached 21%, 24%, and 12%, respectively; when these countries applied [b](closure

**Table 2. Decreasing new case indices of countries by intervention solution.**

| Country | Pre-intervention (% - 95%CI) | Post-intervention (% - 95%CI) | Absolute effect average (AEA) (% - 95%CI) | Relative effect average (% - 95%CI) | Posterior probability of a causal effect |
|---|---|---|---|---|---|
| Spain[No → b,c,d,e] | 25.98 (21.28, 31.10) | 5.92 (4.73, 7.11) | 21 (14, 28) | 78 (51, 106) | 99.995**** |
| Spain[b,c,d,e → b,d,e] | 5.92 (4.73, 7.11) | 1.84 (1.39, 2.41) | 4.3 (2.3, 6.3) | 70 (37, 103) | 99.995**** |
| Italy [No → b,c,d,e] | 25.92 (20.53, 32.48) | 3.86 (3.22, 4.56) | 24 (16, 32) | 87 (58, 117) | 99.995**** |
| United Kingdom [No → b,c] | 22.14 (17.61, 27.23) | 7.53 (6.30, 8.92) | 16 (7.9, 24) | 68 (34, 103) | 99.980*** |
| United Kingdom [b,c → b] | 7.53 (6.30, 8.92) | 3.10 (2.77, 3.46) | 4.6 (2.6, 6.6) | 60 (34, 87) | 99.995**** |
| Canada [No → b,c,e] | 18.70 (15.51, 22.07) | 7.51 (6.46, 8.76) | 10 (5, 15) | 54 (27, 81) | 99.985*** |
| United States [No → b,c,d,e] | 16.28 (12.60, 20.20) | 4.72 (4.01, 5.49) | 12 (5.7, 18) | 72 (34, 110) | 99.995**** |
| France [No → b,c,e] | 18.34 (13.17, 24.14) | 4.01 (2.39, 6.10) | 15 (5.5, 24) | 79 (29, 128) | 99.900** |
| Germany[No → b,c,e] | 15.88 (11.68, 20.48) | 5.36 (4.55, 6.21) | 11 (3.3, 18) | 67 (21, 113) | 99.715** |
| China[b,c,d → a,b,c,d] | 17.29 (12.90, 22.04) | 0.72 (0.52, 0.96) | 16 (10, 23) | 96 (60, 132) | 99.995**** |
| China[a,b,c,d → a,b,d] | 0.72 (0.52, 0.96) | 4.36 (2.62, 7.21) | -3.6 (-3.2, -3.9) | -464 (-508, -418) | 99.995**** |
| China[a,b,d → a,b,e] | 4.36 (2.62, 7.21) | 1,94 (0.85, 3.40) | 2.4 (-1.2, 5.9) | 55 (-29, 136) | 91.001 |
| Korea[No →a,b,c] | 13.98 (9.13, 19.93) | 1.34 (1.12, 1.58) | 14 (6, 21) | 91 (40, 142) | 99.965*** |
| Japan [No → a] | 12.60 (8.30, 17.27) | 7.76 (6.00, 9.58) | 5.8 (-1.2, 13) | 43 (-8.7, 94) | 95.085* |
| Japan[a → a,b,e] | 7.76 (6.00, 9.58) | 9.93 (7.79, 12.07) | -2.2 (-4.7, 0.29) | -29 (-61, 3.7) | 96.025* |
| Japan[a,b,e → a,e] | 9.93 (7.79, 12.07) | 4.56 (2.80, 6.50) | 5.4 (1.9, 9) | 54 (19, 90) | 99.870** |
| Japan[a,e → a,c,d,e] | 4.56 (2.80, 6.50) | 1.63 (1.16, 2.07) | 3.1 (0.46, 5.7) | 65 (9.7, 121) | 98.960* |
| Singapore[a → a,b,c,e] | 11.38 (9.91, 12.96) | 5.09 (4.11, 6.19) | 6.6 (3, 10) | 56 (26, 87) | 99.985*** |
| Hong Kong[a → a,b,c,e] | 8.36 (6.79, 10.01) | 0.67 (0.32, 1.09) | 7.8 (4.8, 11) | 92 (56, 128) | 99.995**** |
| Taiwan[a,b → a] | 4.46 (2.65, 6.59) | 8.93 (6.47, 11.74) | -4.3 (-7.5, -1.2) | -95 (-164, -25) | 99.565* |
| Taiwan[a →a,c,e] | 8.93 (6.47, 11.74) | 1.00 (0.50, 1.68) | 6.6 (3.1, 10) | 87 (41, 133) | 99.97** |

* < 0.05

** < 0.01

*** < 0.001

**** < 0.0001

[a]: centralized isolation of all confirmed cases

[b]: closure of schools

[c]: closure of public areas

[d]: closure of cities

[e]: closure of borders

We analyzed national interventions of each country at the moment of the 10th confirmed cases. The countries applied their control measures before or after this moment independently. As a result, we defaulted that before the arrow "→" was the time before the 10th confirmed cases, and after the '"→" was the time after the 10th confirmed cases; and "No" meant that there was no intervention at that time.

of schools), [c](closure of public places), [d](closure of cities), and [e](closure of borders) (p <0.0001). The AEA of the UK decreased to 16% (p<0.0001) with the closure of their schools[b] and public areas[c]. In Taiwan, there was an increase in AEA to 4.3% (1.2%, 7.5%) after Taiwan reopened their schools. Modification of absolute effect average in each country according to the application of different intervention strategies was depicted in S2 Fig. This result demonstrated that the closure of schools played an essential role in controlling the rise of new cases of COVID-19 in Taiwan.

Based on the implementation of methods[a,b,c,d,e], this study categorized 14 countries into two groups, group A and group B (Fig 1). Group A included Asian countries with strict lockdown policies (Taiwan, Hong Kong, Singapore, Japan, and China) and Asian countries

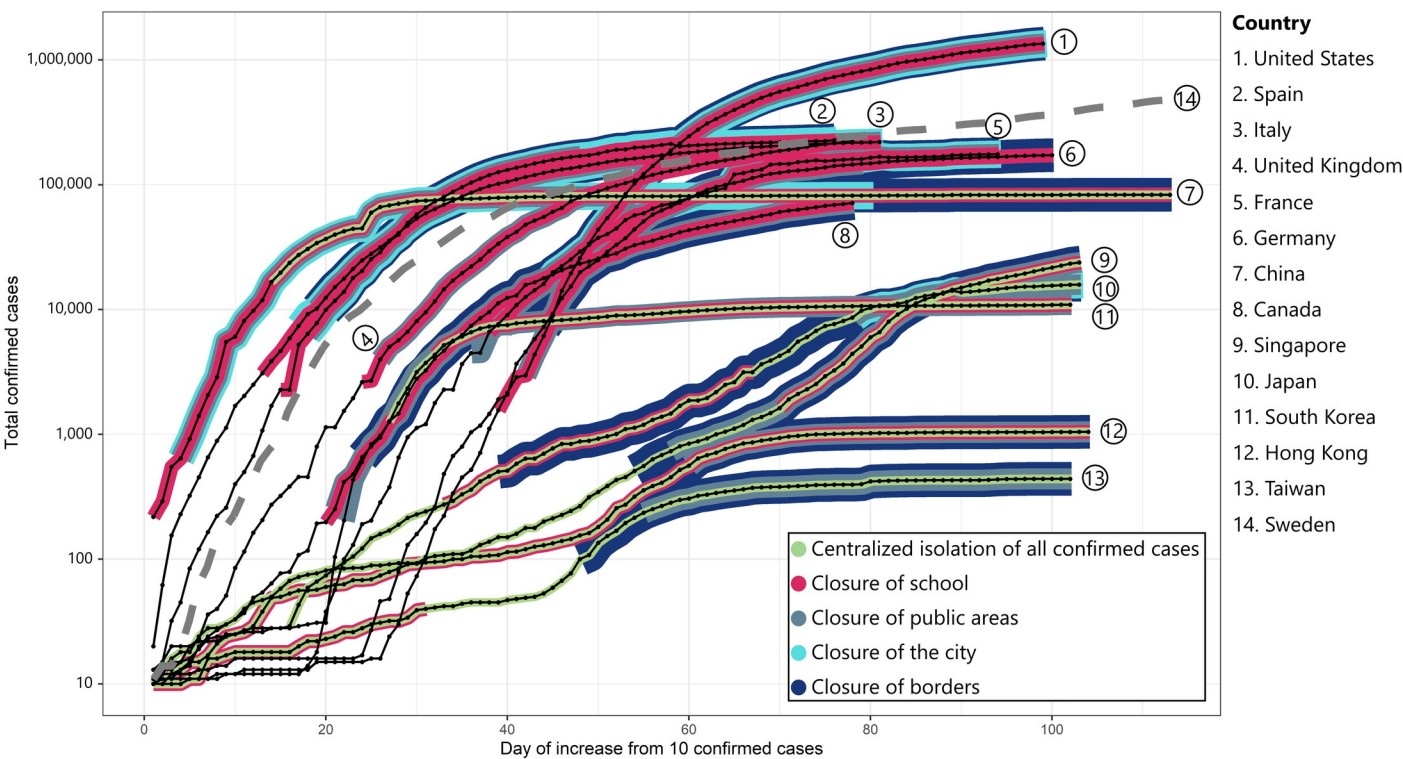

**Fig 1. Analysis of anti-epidemic solutions according to the number of COVID-19 cases trajectories.**

without strict lockdowns (South Korea); they applied method[a]. Group B consisted of European and North American countries with strict lockdown policies (Canada, France, Germany, Spain, Italy, United Kingdom, and the United States) and Sweden; they did not apply method[a]. Group A represented a slow elevation of new cases, as well as a low number of total confirmed cases, regardless of the implementation of the closure of cities and borders, proven by a decrease in new case indices of countries in the group. In this group, Taiwan, Singapore, and Hong Kong isolated all of their confirmed cases[a] before the 10[th] case at 11 days, 8 days, and 4 days, respectively. Next, Japan and South Korea applied methods[a] after their 10[th] cases at 16 days and 27 days, respectively. Exceptionally, China isolated all confirmed cases 14 days after the country's announcement of the outbreak of COVID-19 on January 20, 2020, with 219 confirmed cases. When South Korea and China had their first confirmed cases of COVID-19, the countries did not isolate all of the confirmed cases[a]. This reluctance led to a rapid rise in the number of new cases in these two countries. At this time, new case indices of South Korea and China were on a comparable trend to the countries in group B. However, when the centralized isolation of all confirmed cases[a] was officially implemented on the 14[th] day in China and the 27[th] day in South Korea, the two countries were able to flatten the curve of the confirmed cases of COVID-19 after 13 days of applying method[a] (Fig 1).

In group A, due to the early implementation of centralized isolation of all confirmed cases in Taiwan and Hong Kong, these two countries had lower new case indices compared to Japan, South Korea, and China, which postponed the application of method[a]. According to the analysis, after 60 days since the 10[th] confirmed case, South Korea had 9,661 confirmed cases, whereas this number in Japan, Singapore, Hong Kong, and Taiwan were 1,866 (19.3% compared to South Korea), 844 (8.7% compared to South Korea), 561 (5.8% compared to South Korea), and 306 (3.2% compared to South Korea), respectively. Furthermore, after 100 days

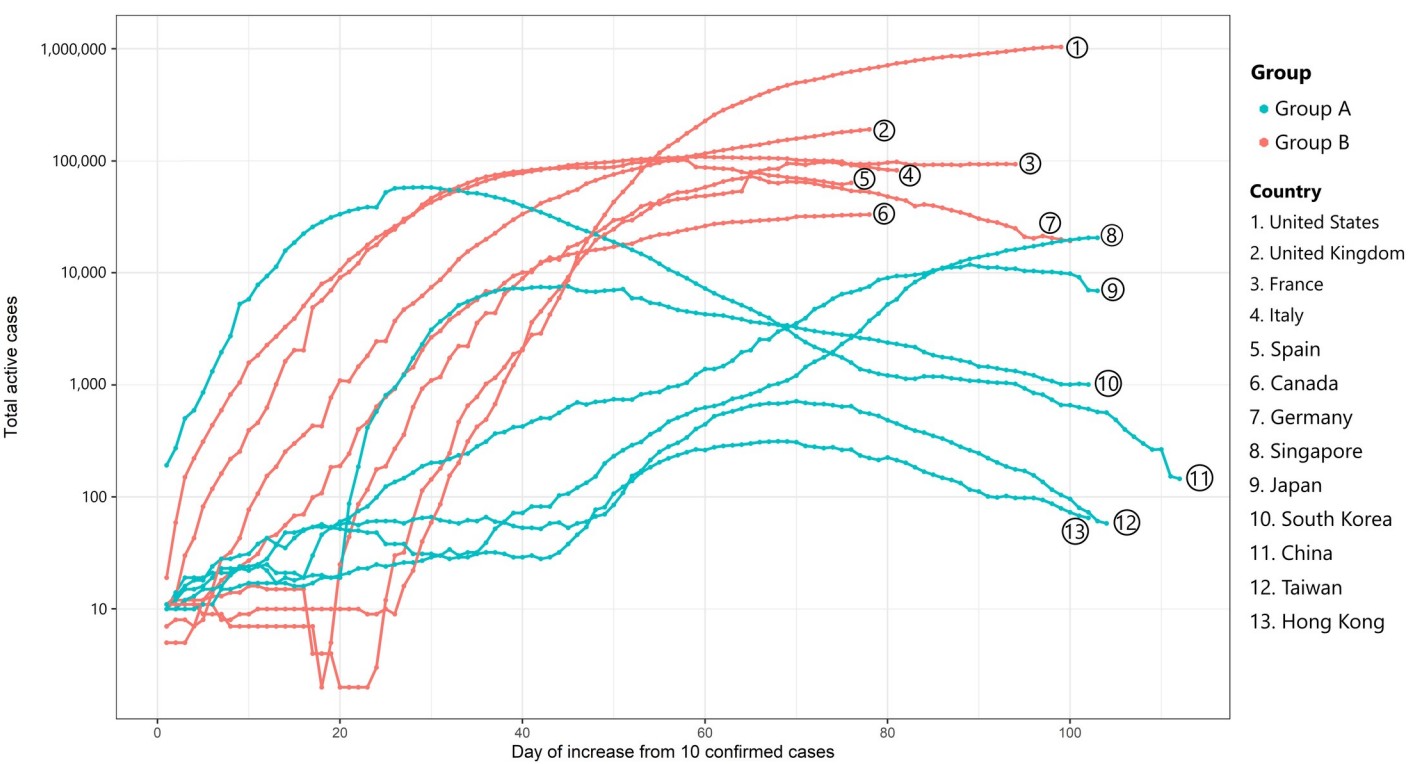

**Fig 2. The comparison of countries' active cases since the 10th confirmed case of COVID-19.**

since the 10[th] confirmed case, Singapore had 21,707 confirmed cases, while this number in Japan, South Korea, Hong Kong, and Taiwan were 15,575 (by 71.8% compared to Singapore), 10,874 (50.0% compared to Singapore), 1,044 (4.8% compared to Singapore), and 440 (2.0% compared to Singapore), respectively (Fig 2). Thus, the earlier that centralized isolation for all confirmed cases was put in place, the more effective the method was at decreasing the new case index of COVID-19. Conversely, in group B, which only applied social distancing measures such as the closure of schools, public areas, and borders without the centralized isolation of all confirmed cases, there were no significant decreases in the number of new cases. Notably, the United States witnessed a high intensity of infection regardless of the closure of these areas. As for Sweden, the number of estimated total confirmed cases (dash line) was comparable to that of countries in group B when the number of new cases has not reached its peak. In which Spain, Italy, the United Kingdom, Germany, and France lessened the day of achieving the highest new cases (day 29[th], day 30[th], day 47[th], day 55[th], and day 65[th], respectively), Sweden reached the peak of cases in day 118[th] with 62,728 confirmed cases according to government reports and 560,895 estimates. The results show that when Sweden did not apply any measures[a,b,c,d,e] and the peak time was 1.8–4.1 times longer than other countries.

On the other hand, the application of centralized isolation would no longer be effective without other solutions such as closing public places, schools, cities, or borders; when the number of active cases was over 400. This was exemplified in Japan. After 40 days since the number of active cases became greater than 400 cases (active cases > 400) in Japan, all methods[a,b,c,d,e] were no longer sufficient to control the rise of new cases, as the active cases increased 19 times with 7,547 active cases (active cases = confirmed cases–death cases–recovered cases). Even when Japan applied methods[a,b,c,d,e] synchronously, after only 14 days since reaching the point of "active cases > 400", the active cases kept rising significantly, in fact, 28 times,

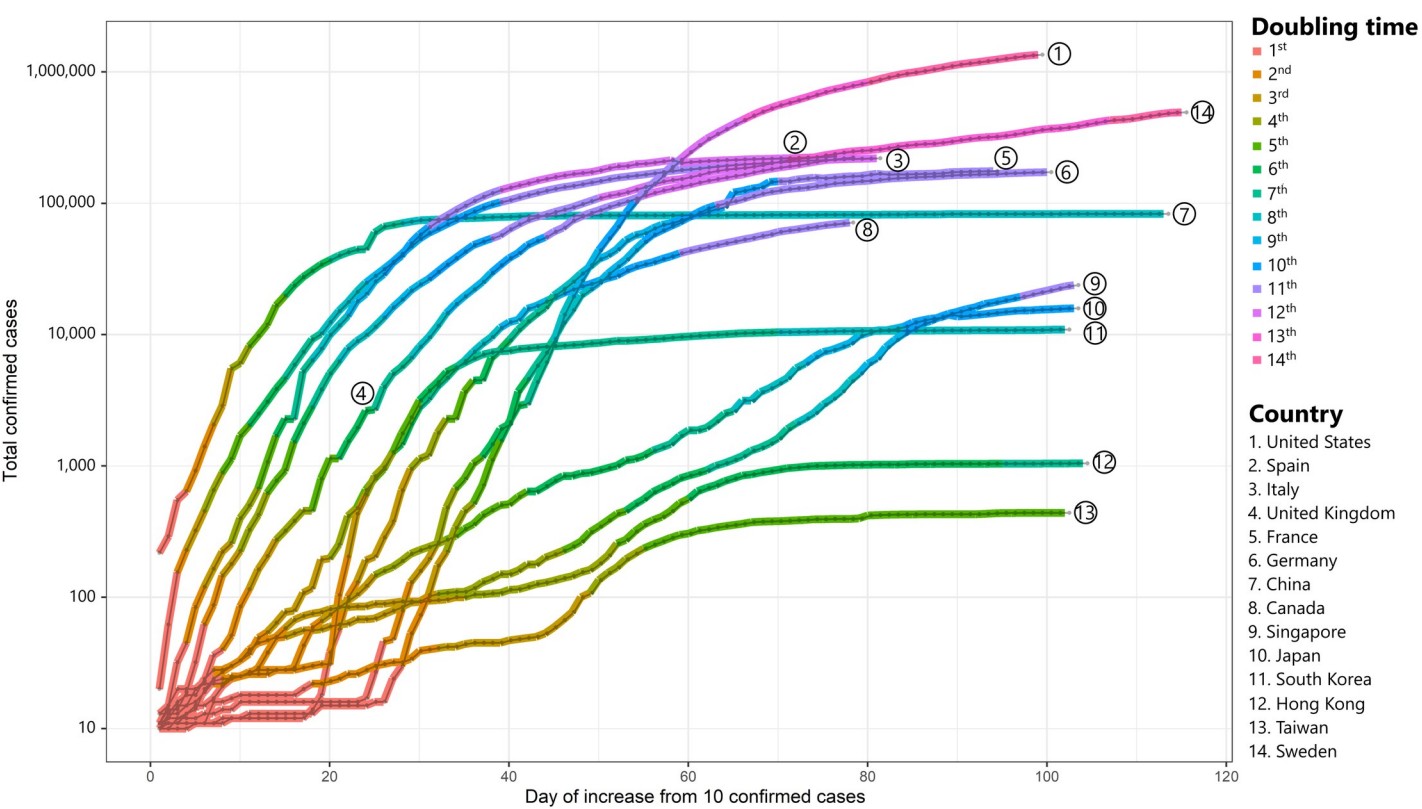

**Fig 3. Doubling time analysis.**

reaching a total of 11,198 active cases. This trend was comparable to Singapore when the country implemented methods[a,b,c,e] after 16 days since reaching the point of "active cases > 400", and the number of active cases tripled to 1,211. Moreover, this number increased rapidly, 52 times, with 20,799 active cases when Singapore also applied methods[a,b,c,e] synchronously after 34 days since reaching the point of "active cases > 400".

In contrast, Hong Kong's active cases increased gradually 1.5 times with 676 active cases, when the government applied methods[a,b,c,e] after 2 days since "active cases > 400". Once again, the data showed that the sooner the countries had measures[a,b,c,d,e] in place, the more sufficiently they could control the pandemic.

Additionally, in order to verify and emphasize our above findings, the two important metrics, such as doubling time, as well as time-varying reproduction number in pre-intervention and post-intervention periods, were also calculated. These two metrics are widely used to evaluate the spread of infectious diseases. The doubling time is the required time to duplicate the number of infected peoples. Given the exponential growth of an epidemic and the constant growth rate r, the doubling time needed is calculated as (ln 2)/r. Thus, an elevation in doubling time suggests a reduction in virus transmission. In our study, the doubling time was calculated by the days that the positive cases increase to double values or the days that the increase of positive cases reaches 100% [12]. Fig 3 demonstrated that the United States suffered the highest doubling time, which was 14 times. In contrast, Taiwan, with its reasonably preventive policies, had the lowest doubling time in our analysis. Indeed, 100 days since 10[th] confirmed cases, group A, which was using the centralized isolation of all confirmed cases, resulted in a lower doubling time: Japan and Singapore's doubling times were 9 and 10, respectively; and the

other four countries in this group had the doubling times under 8. In reverse, the doubling time in group B seems to be more serious, with the excess of the 8th doubling time just after 20–40 days. Notably, the 11th doubling time presented from day 30[th] to day 70[th] since the 10[th] case. In China and South Korea, without centralized isolation of all confirmed cases, their 7[th] doubling times were just after 20[th] day and 34[th] day, respectively, in the pandemic's early stage. However, after applying the aforementioned measure, their doubling times just increased for one more scale after 70 days.

Regarding reproduction rate RR, it evaluates how effective the control measures are in mitigating the growth of the confirmed cases. This index is estimated by the average the population who become infected by one infectious person. Alternatively, it can be calculated through the number of mortality [13]. If RR is under 1, the spread of the virus is slowing. Whereas if the value is greater than 1, the virus spread is increasing [14]. Our study evaluated the effectiveness of intervention through RR before and after applying these solutions. The reproduction rate is obtained from the database of the "Our World In Data" (https://ourworldindata.org/coronavirus-data), a project output of the Oxford Martin Programme on Global Development from the University of Oxford. However, during the early stage of the pandemic, the number of death cases was low, leading to difficulties in estimating the reproduction rate. Therefore, we estimated the missing values by estimating the correlation between the reproduction rate and the ratio of new cases ($day_n$) and new cases ($day_{n-1}$). The time-varying of reproduction rate for all the relevant countries was outlined in Fig 4. On the 13[th] day, countries in group B, which did not apply any control measures, had an average reproduction rate of 2.23 (0.38). However, after applying measures such as the closing of schools, public areas, and borders, this index achieved a reduction from 1.2 to 2.5 points (S8 Table). For countries in group A, the average reproduction rate was 1.28 (0.64) before applying closing public areas, cities, or borders. However, when they used the above measures, this rate fluctuated between 0.093 and 1.1. In testing with the fluctuation from 1 to 4 days, the analysis showed similar statistical results with an insignificant difference.

## Discussion

In this work, we demonstrate that the early implementation of confirmed cases' centralized isolation was the most effective strategy in decreasing the incidence of new COVID-19 cases. Thus, we can deduce that even in the face of an unknown contagion that spreads like wildfire, or one that may not currently have a curative treatment or a vaccine, the lesson to be learned here is that early isolation is key to containment and possibly avoiding a pandemic. However, delaying isolation procedures after confirmed cases have reached more than 400 renders centralized isolation alone insufficient.

Our research was designed to evaluate the efficaciousness of control measures such as the closure of schools, closure of public areas, closure of cities, closure of borders, and centralized isolation of all confirmed cases in mitigating the spread of the pandemic in 14 countries: Canada, China, France, Germany, Hong Kong, Italy, Japan, South Korea, Singapore, Spain, Taiwan, the United Kingdom, the United States, and Sweden. Although these measures implementation varied in each country according to different periods, these solutions were proven to play a crucial role in significantly reducing the spread of COVID-19 after 13 days of application. As governments grapple with how to preserve their general infrastructure, economies and continue classroom education, the data here imparts some valuable wisdom in guiding these decisions. Indeed, centralized isolation of all confirmed cases was determined as a key control measure, as it helped to remarkably restrict the number of new cases after 13 days of application and control the spread of the virus at a low level. However, when the number of

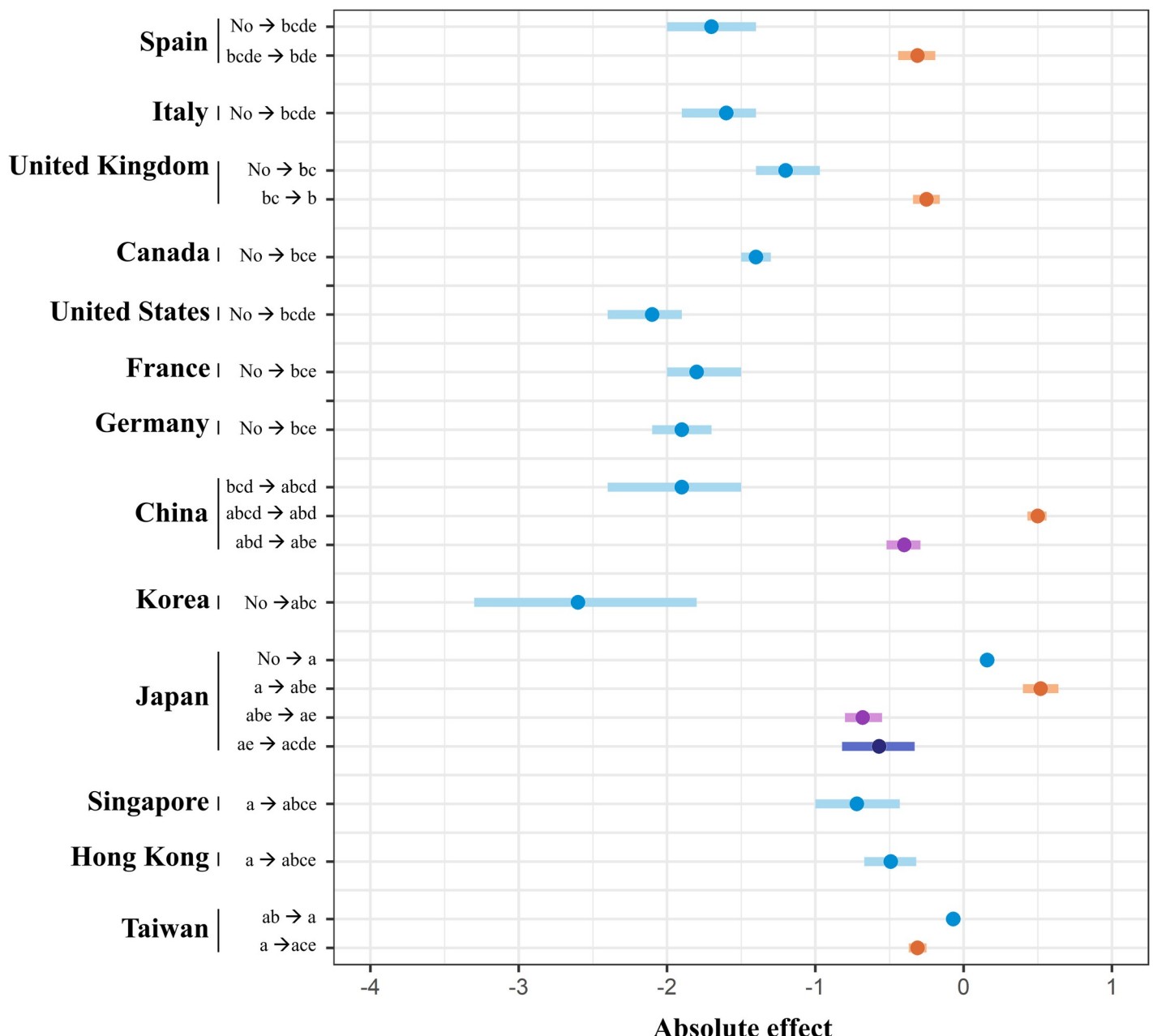

**Fig 4. Absolute effect of government policies on reproduction rate.**

new cases reached over 400 cases, the centralized isolation of all confirmed cases lost its effect if social distancing-related control measures were not simultaneously applied. Particularly, the number of new cases in countries with applications of centralized isolation of all confirmed cases was proven to be lesser than that without the application of this measure. For example, Taiwan opted to reopen its schools before closing public areas. As a result, the absolute effective average (defined as the average percent decrease rate of the trend after intervention/day) increased to 4.3%. This conveys that keeping educational institutions closed is also crucial to decrease the incidence of new COVID-19 cases.

The 14 countries were divided into two groups; group A enlisted the centralized isolation control measure, while group B did not. Interestingly enough, Western countries dominated group B and showed the largest flux in cases (Fig 2). In Fig 1, countries in group A, with the application of centralized isolation of all confirmed cases, represented better management of new confirmed cases than countries in group B. This finding is in line with the study of Zhu et al. [15]. In countries such as China and South Korea that did not apply the centralized isolation of all confirmed cases since the beginning of new cases, the curve flattened after 13 days of applying this measure. In contrast, countries including Singapore, Germany, Japan, Taiwan, and Hong Kong (in group A), showed a distinct low curve of new cases compared to group B.

Currently, while vaccination and approved drugs for COVID-19 require time to assess its efficaciousness, the management of the sources of infection, as well as routes of transmission, remain the vital factors to control the pandemic. However, given the possibility of transmission of SARS-CoV-2 through respiratory droplets [11, 16, 17], air [18, 19], and direct contact with contaminated surfaces [20, 21]; control of the common routes of transmission of SARS-CoV-2 is not sufficient when we let the confirmed cases be self-isolated at home. The rationale behind this is that these individuals continue to do their daily activities such as buying food, entering local pharmacies, and patronizing other businesses, which might result in an increase in the risk of community spread.

The study sets itself apart amidst the sea of data on COVID-19 because it provides a consolidated, systematic comparison of measures utilized by 14 countries significantly impacted by COVID-19 in their efforts to lower the spread of the virus. In the initial months of a rapidly spreading pandemic, governments and healthcare systems alike struggled to harness enough data to approach patient treatment and the public health crisis with acumen. By utilizing the Causal Impact Package, we were able to conduct a study that made randomized controlled trials impossible. This approach, coupled with the Bayesian inference method and Markov Chains Monte Carlo, allowed us to observe the impact of centralized isolation of all confirmed cases, closure of schools, closure of public areas, closure of cities, and closure of borders; on case indices pre-and post-intervention.

Given that our evaluation in Figs 1 and 2 might be subjective, we have performed the cause-effect analysis to assess the control measures' impact on the speed of infection between the current and newly infected people. This method is frequently applied in public health and the economy to estimate the counterfactual and predict what would happen in the absence of a treatment [22, 23]. Our result determined that all non-pharmaceutical interventions, including the closure of schools[b], closure of public areas[c], closure of cities[d], and closure of borders; significantly reduced the trend of the new case index. When all of these social distancing methods were applied, the new case indices varied from 3.86% to 5.92%. When countries applied fewer solutions, this index ranged from 4.01% to 7.53%. In countries without applying these control measures, we revealed a high value of new case indices, from 12.60% to 25.98%. Thus, these social distancing control measures significantly reduced the number of new cases. In the same manner, our findings were in line with the report of Chaudhry et al., which identified that the days to partial or full lockdown and the day to any border closures were the two significant predictors associating with the total number of reported cases per million [24]. Interestingly, this author also figured out that a full lockdown policy was a supporting factor in increasing the number of recovered cases. On the other hand, our study determined a crucial role of centralized isolation of all confirmed cases. In China, the application of centralized isolation of all confirmed cases, but not the closure of borders, also reduced the new case indices to 0.72% (95% CI: 0.52, 0.96). In South Korea, a combination of centralized isolation of all confirmed cases, closure of schools, and closure of public areas decreased the new case indices to 1.34% (95% CI: 1.12, 1,58). A similar outcome was obtained in Hong Kong with a reduction to 0.67%

(95% CI: 0.32, 1.09) with the application of centralized isolation of all confirmed cases and closure of borders, schools, and public areas.

Isolation strategies were key in displacing the upward trend of infection rates due to the human-to-human spread of COVID-19 being the mechanism of rising new cases. After only 13 days of applying the centralized isolation method, despite failing to initially isolate all confirmed cases, as demonstrated in Fig 1, South Korea and China could flatten the confirmed cases' curve via centralized isolation. Furthermore, Taiwan and Hong Kong had even lower new case indices because they put centralized isolation into effect early. However, a potential limitation could be if cases exceeded the 1000 threshold in the early stages upon discovering a pneumonia-like illness before centralized isolation could be implemented. At that juncture, the immediate move would be to enlist other measures such as social distancing, wearing masks, closure of schools, and closure of public spaces. Of note, confirmed cases of >400 with all methods implemented still demonstrated a trend of increasing new cases. Therefore, it again brings us back to the evidence fleshed out by the data; early implementation of centralized isolation when confirmed cases are low (<100) is imperative to controlling the spread. History has shown that a two-step with the Coronavirus family is not a singular battle of the SARS outbreak in Guangdong, China in 2002 [6] and the MERS outbreak in Saudi Arabia in 2012 [7]. Hence, one of the main takeaways here is there is an urgent need to redesign government and healthcare systems-level protocols in response to pneumonia-like illnesses and (or) those that are suspected of spreading via respiratory droplets.

Without a doubt, early and rapid action by governments and healthcare workers to isolate confirmed cases of COVID-19 played a key role in lowering the case indices for countries that implementing the strategies before reaching the threshold that required additive measures. However, as human nature dictates, adherence to isolation and other strategies, e.g., social distancing and wearing masks, can be difficult when not strictly enforced. Thus protocols for enhancing community compliance with the secondary measures to contain the spread further must be studied.

Despite careful preparation, we acknowledged some limitations in our study that need to be identified. At first, the assumed period of 14 days might be a senseful approximation since the time between the onset of symptoms and testing is hardly detectable. Even though our study demonstrated the crucial role of centralized isolation in fighting the pandemic and may make other measures superfluous, but only if this measure is imposed at an early stage of virus spread, we also face a current challenge of identification all infections at an early stage of the outbreak. Detection of asymptomatic but infected people who had never been tested represented as an important matter that needed to be discussed. Besides, Asian countries with smaller territory sizes and accumulated gaining experiences of SAR-CoV-1 in the past had a greater benefit to tackle the spread of this infectious disease than European and American countries. Furthermore, Asian countries with "authoritarian government" and "collectivist culture" that pushed societal interests to the forefront might be easily adapted the lockdown policy. In contrast, European and American countries where "individualistic culture" is dominant are challenging to comply with strict enforcement of regulations. Finally, the heterogeneity in the analysis is unavoidable due to difficulties to reach a comparable definition of the time for each measure in countries such as the United States, Canada, and China. Also, there were differences in testing policies and the criteria of confirmed cases in each country.

## Conclusion

Our analysis determined that early centralized isolation was represented as a core intervention in significantly disrupting the pandemic's spread, especially in South Korea. This strategy also

helped Taiwan, Hong Kong, Singapore, and Japan in successfully controlling the crisis in the early stage, when the total number of cases were under 100, without the requirement of the closure of cities and public areas; which could impose a negative impact on the society and economy. However, when the number of cases increased with the apparition of new clusters, coordination between centralized isolation and non-pharmaceutical interventions would facilitate controlling the crisis efficiently.

## Supporting information

**S1 Fig. Change in new case index according to the application of different intervention strategies.**
(TIF)

**S2 Fig. Modification of absolute effect average in each country according to the application of different intervention strategies.** (Posterior probability of a causal effect; * < 0.05, ** < 0.01, *** < 0.001, **** < 0.0001).
(TIF)

**S1 Table. 13 countries with the highest percentage of days having the highest number of daily confirmed cases.**
(DOCX)

**S2 Table. The average value of the ratio of daily new case between the 5 countries and Sweden.**
(DOCX)

**S3 Table. Level of lock-down strategy and its definition.**
(DOCX)

**S4 Table. Downtime of the lock-down strategy as of May 11.**
(DOCX)

**S5 Table. Details of the date of the control measures in each city and state of the United States, Canada, and China.**
(XLSX)

**S6 Table. Insignificant data changes due to the fluctuation of time between testing and reporting in official statistics.**
(DOCX)

**S7 Table. Reference link of the date of implementation of the national intervention at the moment of the 10th confirmed cases.**
(XLSX)

**S8 Table. Absolute effect of government policies on reproduction rate with different timelines.**
(DOCX)

## Author Contributions

**Conceptualization:** Nguyen Hai Nam, Nguyen Tien Huy.

**Data curation:** Nguyen Hai Nam, Phan Thi My Tien, Le Van Truong, Toka Aziz El-Ramly.

**Formal analysis:** Nguyen Hai Nam, Phan Thi My Tien, Le Van Truong, Toka Aziz El-Ramly, Pham Gia Anh, Nguyen Thi Hien, El Marabea Mahmoud, Mennatullah Mohamed Eltaras,

Sarah Abd Elaziz Khader, Mohammed Salah Desokey, Ramy Magdy Gayed, Shamael Thabit Mohammed Alhady, Bao-Tran Do Le, Do Phuc Nhu Nguyen, Ranjit Tiwari, Mohammed Eldoadoa, Britney Howard.

**Funding acquisition:** Phan Thi My Tien.

**Investigation:** Nguyen Hai Nam, Le Van Truong.

**Methodology:** Nguyen Hai Nam, Le Van Truong, Tran Thanh Trung.

**Project administration:** Nguyen Hai Nam, Phan Thi My Tien, Toka Aziz El-Ramly, Nguyen Thi Hien.

**Resources:** Nguyen Hai Nam, Le Van Truong, Pham Gia Anh.

**Software:** Nguyen Hai Nam, Phan Thi My Tien.

**Supervision:** Nguyen Hai Nam.

**Validation:** Nguyen Hai Nam, Phan Thi My Tien, Le Van Truong, Toka Aziz El-Ramly.

**Visualization:** Nguyen Hai Nam, Le Van Truong.

**Writing – original draft:** Nguyen Hai Nam, Phan Thi My Tien, Le Van Truong, Toka Aziz El-Ramly, Pham Gia Anh, Nguyen Thi Hien, El Marabea Mahmoud, Mennatullah Mohamed Eltaras, Sarah Abd Elaziz Khader, Mohammed Salah Desokey, Ramy Magdy Gayed, Shamael Thabit Mohammed Alhady, Bao-Tran Do Le, Do Phuc Nhu Nguyen, Ranjit Tiwari, Mohammed Eldoadoa, Britney Howard, Tran Thanh Trung.

**Writing – review & editing:** Nguyen Tien Huy.

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
