## [Decision Letter · Decision Letter 0]

4 Jan 2021

PONE-D-20-33060

Early centralized isolation strategy for all confirmed cases of COVID-19 remains a core intervention to disrupt the pandemic spreading significantly

PLOS ONE

Dear Dr. Huy,

Thank you for submitting your manuscript to PLOS ONE. After careful consideration, we feel that it has merit but does not fully meet PLOS ONE’s publication criteria as it currently stands. Therefore, we invite you to submit a revised version of the manuscript that addresses the points raised during the review process.

In particular, please pay attention to the following comments, which are in my opinion the most important:

While both reviewers are happy that 13 countries and regions are compared in the manuscript, they also request that you explain why these 13 countries are selected, instead of others;The authors should provide references (like government announcements) that describe the control measures in greater detail. Reviewer #2 also suggested a rough classification of the 13 countries selected into three categories: (a) Asian countries with strict lockdowns, (b) Asian countries without strict lockdowns, and (c) European and North American countries with strict lockdowns, and also using Sweden as an European country without lockdowns as a control.How the effectiveness of lockdowns would be affected by the delay between the time of infection and isolation. For your consideration, Reviewer #2 broke down this delay into three components: (a) the time between infection and onset of symptoms, (b) the onset of symptoms and the time of testing, and (c) the time of testing and reporting in official statistics.Reviewer #1 would like to see more discussions on the limitations of this study, so as to guide future studies.

We look forward to receiving your revised manuscript.

Kind regards,

Siew Ann Cheong, Ph.D.

Academic Editor

PLOS ONE

Journal Requirements:

2. Please ensure links to all stated data sources have been included or cited in the methods section.

"The funders had no role in study design, data collection and analysis, decision to publish, or preparation of the manuscript,"

Reviewers' comments:

Reviewer's Responses to Questions

**Comments to the Author**

1. Is the manuscript technically sound, and do the data support the conclusions?

Reviewer #1: Partly

Reviewer #2: Yes

2. Has the statistical analysis been performed appropriately and rigorously? 

Reviewer #1: I Don't Know

Reviewer #2: Yes

3. Have the authors made all data underlying the findings in their manuscript fully available?

Reviewer #1: No

Reviewer #2: Yes

4. Is the manuscript presented in an intelligible fashion and written in standard English?

Reviewer #1: Yes

Reviewer #2: Yes

5. Review Comments to the Author

Reviewer #1: The article addresses a relevant and current topic. Explains the scientific background and rationale of the investigation. The methods section describes the setting, locations and relevant dates. The criteria of selection of countries were described. However, some elements could be better described. Data from the World Health Organization (March, 15) demonstrated that other countries have more cases or mortality, especially in Europe. Thus, it is important to explain why these countries were not selected. Data relative to control measures could be better described. The documents used as a reference to identify control measures must be referenced in the text for readers to consult. It is necessary to revise these references. Data from Spain (table 1) for example point to the closure of schools on March 11 . However, this only occurred in the entire country after March 14. The same situation applies to control relaxation measures. In Spain, the control reduction plan is published on April 28, 2020 with different phases in each autonomous community. Another aspect in relation to control measures is the difficulty of establishing a date and intensity of measures in continental countries such as China, Canada and the United States. I highlight the situation in the USA, where each state institutes different measures. This question should be clearer in the methods, results tables and limitations of the Study. Another relevant aspect is that it would be important to describe what was considered centralized isolation of all confirmed cases in the method section. Guidelines recommend isolation for all cases in most countries (including the WHO guideline). Thus, it is important for reads understand how the researchers define the difference between countries and references documents (the isolation were compulsory?). Other factors to understand the control measures like testing strategies were not described, analysed in the method section or mentioned in the discussion section. This aspect is important because a country could define isolation of all cases and do not test their citizens adequately, for example, and do not isolate a great part of the infected citizens. The statistics analysis must be described better in a supplementary document or in the text. The results were well described, however it could be necessary to correct some results (changes in the definition of date of implementation of the national intervention and analysis of other countries) . The discussion section could explore other hypothesis and aspects that could justify differences. All countries in group A were European or Americans with occidental culture. The group B has only Asiatic regions/countries. The experience with SAR-COV 1 was not mentioned like one aspect of difference between Asiatic and European and American Countries results. Different territory sizes and governance differences were not mentioned too (Asiatic regions were smaller and some are islands - these are importants factors in epidemic control). It could be important to describe the differences. The study limitations could be better described (difficulties to defines the time of each measure, cases criteria definitions in which country, testing policies, governance differences etc). The conclusions were well described and were adequate to the results.

Reviewer #2: Review

The authors attempt to assess the effectiveness of nonpharmaceutical interventions in fighting the SARS-CoV-2 pandemic. The study incorporates 13 countries and compares their strategies. From the methodological point of view, infections are analysed over time, whilst testing the influence of different measures on the temporal development of infections. The authors conclude that centralized isolation of confirmed infected individuals is an effective strategy (on condition that is implemented early) rather than “lockdowns”. The study is very interesting and of high quality from a methodological perspective. The simultaneous comparison of different countries (with different economic and cultural background) is to be welcomed. However, there are some issues I would recommend to address in a revision (“major revisions”), especially with respect to the selection of countries and the interpretation of the results (see below).

MAJOR COMMENTS

1) Again, I have to express my happiness about the comparison of different countries with different containment strategies. However, there is an important issue regarding the selection of these countries: With respect to the “lockdown” policy, the selected countries in the study can be divided into three groups (not to be confused with the classification made by the authors on page 8): A) Asian countries WITH strict “lockdowns”, B) Asian countries WITHOUT strict “lockdowns” and C) European and North American countries WITH strict “lockdowns”. However, there is one North European country (Sweden) which did NOT impose a “lockdown” as well (Note: We have to keep in mind that there have been several measures against SARS-CoV-2 spread both mandatory and voluntary – it is a common “fairy tale” that Sweden did nothing against the virus! Furthermore, assessing the Sweden strategy is complicated as, in the first wave of infections, Sweden had a much higher COVID mortality than its neighbours but a lower mortality compared to France, Italy, or Belgium with hard “lockdown” measures). But I would strongly recommend including Sweden into the analysis as a control group (European country WITHOUT “lockdown” AND without centralized isolation).

See this paper on Sweden by Born et al. 2020: https://cepr.org/sites/default/files/news/CovidEconomics16.pdf#page=6.

2) As the analysis incorporates infection/fatalities data for 13 countries, it is reasonable to use a data source which provides all this case data for each country. Like in many studies towards the effectiveness of nonpharmaceutical interventions, the database from the Johns Hopkins University (JHU) was used. This data does, of course, not contain information about the true infection date (which is unknown in the majority of cases). By consequence, valid analyses of the effectiveness of nonpharmaceutical interventions must incorporate an appropriate delay between infection an the (possible) impact of measures. In the present study, the authors consider this problem by referring to the 95% percentile of the SARS-CoV-2/COVID-19 incubation period from the literature. The assumed period of 13 days might be a senseful approximation. However, this delay consists of (at least) three time periods: A) time between infection and onset of symptoms (if any; incubation period), B) time between onset of symptoms (if any) and testing, C) time between testing and reporting in official statistics (including JHU database). Note that these time periods may overlap. The authors should clarify this and should comment how this assumption on the delay might affect the results.

For further information on the issue of reporting delays etc., see for example the following literature:

Flaxman et al 2020 (https://doi.org/10.1038/s41586-020-2405-7) and Homburg 2020 (https://doi.org/10.1515/ev-2020-0010) also use JHU data for international comparisons of NPI effectiveness, but they trace infection dates back from reported death cases. A similar method is used by Wood 2020 (https://arxiv.org/pdf/2005.02090.pdf). A critique on former NPI studies using the JHU data (explicitly discussing the “reporting delay problem”) can be found in Wieland 2020 (https://doi.org/10.1016/j.ssci.2020.104924).

3) There is an issue with respect to the interpretation of the effectiveness of CENTRALIZED isolation: “[…] is not sufficient when we let the confirmed cases be self-isolated at home. The rationale behind this is that these individuals continue to do their daily activities such as buying food, entering local pharmacies, and patronizing other businesses, which might result in an increase in the risk of community spread.” (page 11, lines 323-326). I cannot assess the practical implementation of isolation in Singapore or Taiwan but, in European countries, domestic quarantine means a strict stay-at-home order which is controlled by health authorities. Is the centralized isolation strategy efficient because it is CENTRALIZED or because other types of isolation (such as domestic quarantine) are not or cannot be controlled? This is a more general issue, as, in the “Discussion” section, measures are supported because of their rigour: “However, as human nature dictates, adherence to isolation and other strategies e.g. social distancing and wearing masks, can be difficult when not strictly enforced. Thus protocols for enhancing community compliance with the secondary measures to further contain the spread, must be studied” (page 13, lines 375-378). How does this conclusion match the results from several European countries with differing NPIs and differing levels of self-responsibility? See e.g. point 1 with respect to Sweden. I would not agree that any measure should be on a volunteer basis, but, in democratic states under the rule of law, governments must count on the support of individuals. I would ask the authors to include these aspects in the discussion, because voluntary compliance to (reasonable) rules are an important foundation of democratic societies.

MINOR COMMENTS

4) The criteria for county selection appear reasonable in most cases, but please describe (just 1-2 sentences) what is meant by “having a close relationship with the outbreak's origin (Wuhan, China)”.

5) Centralized isolation is identified as an effective strategy in fighting the pandemic and may make other measures superfluous, but only if this measure is imposed at an early stage of virus spread (“On the other hand, the application of centralized isolation would no longer be effective without other solutions such as closing public places, schools, cities, or borders; when the number of active cases was over 400”; page 9, lines 265-267). We know that SARS-CoV-2 cases are underestimated everywhere because of an unknown “dark figure” of asymptomatic but infected people who have never been tested. However, official reported cases are the only data source available, and, according the WHO, “asymptomatically infected individuals are much less likely to transmit the virus than those who develop symptoms” (http://www.emro.who.int/health-topics/corona-virus/transmission-of-covid-19-by-asymptomatic-cases.html). Is it realistic to identify all infections at an early stage of the pandemic? I would recommend that the authors should reflect this issue, just in 1-2 sentences.

6) Page 6, line 156: “Since vaccines and target drugs are not available…” should be more like “Since vaccines and target drugs were not available in March 2020…”. A similar issue is on page 11, line 319: “…while vaccination and approved drugs for COVID-19 are still unavailable…”. Note that I received the review invitation on December 12, 2020, and I don’t know when the manuscript was written and submitted. Now, there is a vaccine.

7) The authors should refer some previous literature towards analyses of the effectiveness of NPIs, especially country comparisons. See the literature in point 2 and, e.g., Chaudry et al 2020 (https://doi.org/10.1016/j.eclinm.2020.100464).

8) On page 13, lines 385-386, the authors write that “lockdown” policies have “a negative impact on the society and economy”. Like the majority of scientists, I would agree with that. However, it may be a good idea to cite some literature on negative impacts of “lockdowns” already in the “Introduction” section.

For example, Miles et al. 2020 (https://doi.org/10.1111/ijcp.13674) balance the costs and benefits of “lockdowns” and find that the economic impacts are more harmful than their benefits (economic consequences). Adams-Prassl et al 2020 (https://hceconomics.uchicago.edu/research/working-paper/impact-coronavirus-lockdown-mental-health-evidence-us) identify severe impacts of stay-at-home orders on mental health (psychosocial consequences).

9) Of course, it’s a matter of style but maybe the authors should use the term “nonpharmaceutical interventions” at least once in the text and/or the abstract (because this is the “de facto official” term for containment measures in the international literature). But, keep in mind that this is a “style question” :)

10) The figures are a little bit blurred in the review version, but this may be due to graphics compression during the processing of the manuscript?!

FURTHER NOTES

As I am not an English native speaker, I cannot assess language issues properly.

Point 3: “Have the authors made all data underlying the findings in their manuscript fully available?” – The authors declared: “Yes - all data are fully available without restriction”. As a reviewer, I have no access to this data.

6. PLOS authors have the option to publish the peer review history of their article (what does this mean?). If published, this will include your full peer review and any attached files.

Reviewer #1: **Yes: **Alexandre Medeiros de Figueiredo

Reviewer #2: **Yes: **Thomas Wieland

---

## [Author Response · Author response to Decision Letter 0]

21 Feb 2021

February 18th, 2021

PLOS ONE PONE-D-20-33060

Early centralized isolation strategy for all confirmed cases of COVID-19 remains a core intervention to disrupt the pandemic spreading significantly

Dear Editors and Reviewers,

We are most grateful to the Editor, the Associate Editor, and the Reviewer for providing helpful and constructive comments regarding our manuscript. We have taken all of these comments into account and hereby submit a revised manuscript with the changes underlined.

We responded to all of the comments as indicated below, and we hope that our explanations and revisions will be deemed satisfactory.

Sincerely,

Associate Nguyen Tien Huy, M.D, Ph.D.

School of Tropical Medicine and Global Health, Nagasaki University, Nagasaki 852-8523, Japan 

E-mail: tienhuy@nagasaki-u.ac.jp

 

The comments of Academic Editor’s Siew Ann Cheong to the Author:

After careful consideration, we feel that it has merit but does not fully meet PLOS ONE’s publication criteria as it currently stands. Therefore, we invite you to submit a revised version of the manuscript that addresses the points raised during the review process.

In particular, please pay attention to the following comments, which are in my opinion the most important:

Comment 1) While both reviewers are happy that 13 countries and regions are compared in the manuscript, they also request that you explain why these 13 countries are selected, instead of others.

Author Response: Dear Editor, we are thankful for your comment, and we are pleased to answer your question. From 70 days from January, 22nd 2020 (when the John Hopkins Coronavirus Resource Center began to collect the data of COVID-19) to the end of March 2020, we have collected data of countries with the highest number of daily confirmed cases. Among them, United States, Canada, and China, with the highest populations, were chosen due to their representative characteristic. Afterward, 5 Asian countries and 5 European countries with the percentage of days having the highest number of daily confirmed cases higher than 60% were chosen for later analysis. Additionally, Sweden, a North European country that decided to live in peace with the virus, was chosen as a European control country without both lockdown and centralized isolation. Details of the percentage of days having the highest number of daily confirmed cases of 13 countries were outlined in Supplementary table 1 

Text Insertion (if applicable)/ page/ line number of change:

(Page 4; line 100-109) 

“In order to assess the impact of current control measures on the number of new cases, during 70 days from 22 January 2020 (when John Hopkins Coronavirus Resource Center began to collect the data of COVID-19) to the end of March 2020, we have collected data of countries that had the highest number of daily confirmed cases. The United States, Canada, and China, with the highest population, were chosen due to their representative characteristic. Afterward, 5 Asian countries and 5 European countries with the percentage of days having the highest number of daily confirmed cases higher than 60% were chosen for later analysis. Additionally, Sweden, a North European country that decided to live in peace with the virus, was chosen as a European control country without both lockdown and centralized isolation. Details of the percentage of days having the highest number of daily confirmed cases of 13 countries were outlined in the S1 Table.”

S1 Table: 13 countries with the highest percentage of days having the highest number of daily confirmed cases

ID Country N (days) Percentage of days having the highest number of daily confirmed cases

1 China 70 100.00%

2 South Korea 70 100.00%

3 United State 70 100.00%

4 France 68 97.14%

5 Germany 65 92.86%

6 Italy 61 87.14%

7 United Kingdom 61 87.14%

8 Japan 60 85.71%

9 Canada 53 75.71%

10 Singapore 51 72.86%

11 Hong Kong 48 68.57%

12 Spain 47 67.14%

13 Taiwan 43 61.43%

Comment 2) The authors should provide references (like government announcements) that describe the control measures in greater detail. 

Author Response: Dear Editor, thank you so much for your comment. An additional paragraph and supplementary table 5 (attached excel file) had been added to outline the detail of each control measures as below: 

Text Insertion (if applicable)/ page/ line number of change:

(Page 5-6; line 135-159) 

“In China, Taiwan, and Hong Kong, all suspected and confirmed cases were isolated in the healthcare facilities and could be ruled out after a negative of SARS-CoV-2 tests with at least 24-hour intervals regardless of the days of isolation. In South Korea and Singapore, a mandatory of 14 days of isolation was applied for all suspected and confirmed cases of COVID-19. In Japan, only suspected and confirmed cases with related symptoms were isolated at the hospitals, whereas the remaining cases without any symptoms were strictly quarantined at home. Even though the policy of centralized isolation of all confirmed cases was modified across these six countries, we assumed these control measures were comparable applied. Thus, centralized isolation of all confirmed cases is defined as a quarantine of all confirmed cases in hospitals, medical camps, or hotels under the supervision of medical staff and during a required time. Closure of schools was applied with the suspension of all cultural, educational, sports, and teaching activities. Closure of public areas consisted of non-essential business closures such as stores, shopping centers, services, such as bars, restaurants, cinemas, theaters, and prohibition of mass gathering. Additionally, citizens were requested not to leave the house unless for essential activities, always keep a safe distance (for example, at least 6-feet) and work from home. Closure of cities involved the restrictions on travel with the suspension of all forms of transportation across cities. Closure of borders is defined as the official announcement of the governments regarding the prohibition of entry and exit of all individuals from all entries, including lands, seas, rails, and air routes. Particularly, to address the heterogeneity of the date of implementation of control measures in countries with various states and cities such as the United States, Canada, and China, we assumed that the first date of applying the lockdown strategies is the representative date for these countries. Details of the date of the control measures mentioned above in each city and state of the United States, Canada, and China with reliable references were summarized in S5 table.”

Comment 3) Reviewer #2 also suggested a rough classification of the 13 countries selected into three categories: (a) Asian countries with strict lockdowns, (b) Asian countries without strict lockdowns, and (c) European and North American countries with strict lockdowns, and also using Sweden as an European country without lockdowns as a control.

Author Response: Dear Editor, thank you so much for your comment. Using Sweden as a control country have helped to reinforce and demonstrate the importance of the implementation of the five control measures such as (a) centralized isolation of all confirmed cases, (b) closure of schools, (c) closure of public areas, (d) closure of cities, and (e) closure of borders. Our analysis showed that when other countries began to start the preventive methods, Sweden, without any implementation of control measures, continued to increase the number of new cases. While Spain, Italy, the United Kingdom, Germany, and France lessened the day of achieving the highest new cases (day 29th, day 30th, day 47th, day 55th, and day 65th, respectively), Sweden reached the peak of cases in day 118th with 62,728 confirmed cases according to government reports and 560,895 estimates. Additional analysis, writing, and update of figure 1 were added according to the recommendations and comments of the Reviewer as below:

Text Insertion (if applicable)/ page/ line number of change:

(Page 4; line 91-94) 

“Thus, our study made an effort to summarize and highlight distinctive features of significant control measures among 14 particular countries, including China, Hong Kong, Taiwan, Singapore, Korea, Japan, the United States, France, Germany, the United Kingdom, Canada, Italy, Spain, and Sweden.”

(Page 4; line 106-108) 

“Additionally, Sweden, a North European country that decided to live in peace with the virus, was chosen as a European control country without both lockdown and centralized isolation.”

(Page 4-5; line 115-124) 

“Regarding Sweden, since the testing policy primarily focused on citizens with symptoms associating with COVID-19 infection and requirement of the inpatient hospital care and (or) elderly care, whereas people with mild symptoms were skipped for contacting the healthcare[3], the number of new confirmed cases of Sweden reported from JHU might be lower than the actual new confirmed cases in reality. To address this limitation, we have estimated the number of new confirmed cases in Sweden by using the number of new confirmed cases of neighboured European countries such as Spain, Italy, the United Kingdom, France, and Germany at the moment without application of any control measures. Our analysis revealed that the actual new confirmed cases of Sweden, in reality, was 8.47 times higher than the report of JHU (S2 Table).”

(Page 8; line 234-235) 

“Sweden was excluded from all of the above control measures since this country denied to against the virus (Table 1).”

(Page 13; line 265-270) 

“Based on the implementation of methodsa,b,c,d,e, this study categorized 14 countries into two groups, group A and group B (Figure 1). Group A included Asian countries with strict lockdown policies (Taiwan, Hong Kong, Singapore, Japan, and China) and Asian countries without strict lockdowns (South Korea); they applied methoda. Group B consisted of European and North American countries with strict lockdown policies (Canada, France, Germany, Spain, Italy, United Kingdom, and the United States) and Sweden; they did not apply methoda”

(Page 13, lune 286-287) 

Figure 1: Analysis of anti-epidemic solutions according to the number of COVID-19 case trajectories.

(Page 14; line 304-310) 

As for Sweden, the number of estimated total confirmed cases (dash line) was quite comparable to that of countries in group B when the number of new cases has not reached its peak. In which Spain, Italy, the United Kingdom, Germany, and France lessened the day of achieving the highest new cases (day 29th, day 30th, day 47th, day 55th, and day 65th, respectively), Sweden reached the peak of cases in day 118th with 62,728 confirmed cases according to government reports and 560,895 estimates.. The results show that when Sweden did not apply any measuresa,b,c,d,e and the peak time was 1.8 – 4.1 times longer than other countries.

(Page 21, line 519-520) 

S2 Table: The average value of the ratio of daily new cases between the 5 countries and Sweden.

Comment 4) How the effectiveness of lockdowns would be affected by the delay between the time of infection and isolation. For your consideration, Reviewer #2 broke down this delay into three components: (a) the time between infection and onset of symptoms, (b) the onset of symptoms and the time of testing, and (c) the time of testing and reporting in official statistics.

Author Response: Dear Editor, we are thankful for your notice and completely agree with the declaration that the assumed period of 13 days might be a senseful approximation, and this delay consists of (at least) three time periods: A) time between infection and onset of symptoms (if any; incubation period), B) time between the onset of symptoms (if any) and testing, C) time between testing and reporting in official statistics. To solve this issue, we conducted the study with the hypothesis that the time between testing and reporting in official statistics could be fluctuating from one day to four days with the details were described in supplementary table 6. After the analysis, the results showed that the data changes due to the fluctuation of time between testing and reporting in official statistics led to insignificant differences of 0-3.5% (CI 95%). Thus, we decided to select day 13th as a time point for analysis. An additional paragraph in the method section and in the limitation section of the discussion as well as supplementary table 6 have been added to clarify this matter.

Text Insertion (if applicable)/ page/ line number of change:

(Page 7; line 194-207)

“From the literature, the average incubation period for SARS-Cov-2 is 5.2 days with the 95% confidence interval of the distribution at 13 days8. However, we encountered a delay in the time reporting of data compared to the testing time in reality. There were some possible reasons for this. Firstly, there were 13 days of average incubation period as mentioned above. Secondly, the differences in time zones between 14 countries could lead to the differences in reporting the data to the public between JHU and these countries. Besides, from April 2020, the JHU database only updated data at the constant time from 3:30-4:00 pm (UTC). Thirdly, there was a lack of simultaneousness in reporting confirmed cases, recovered cases, and death cases of COVID-19. To solve this issue, we conducted the study with the hypothesis that the time between testing and reporting in official statistics could be fluctuating from one day to four days with the details were described in S6 table. After the analysis, the results showed that the data changes due to the fluctuation of time between testing and reporting in official statistics led to insignificant differences of 0-3.5% (CI 95%). Thus, we decided to select day 13th as a time point for analysis.” 

(Page 18; line 431-433) 

Despite careful preparation, we acknowledged some limitations in our study that need to be identified. At first, the assumed period of 14 days might be a senseful approximation since the time between the onset of symptoms and testing is hardly detectable. 

(Page 21, line 525-526) 

S6 Table: Insignificant data changes due to the fluctuation of time between testing and reporting in official statistics

Comment 5) Reviewer #1 would like to see more discussions on the limitations of this study, so as to guide future studies.

Author Response: Dear Editor, thank you so much for your comment. As your suggestion, we have updated the limitation sections as below:

Text Insertion (if applicable)/ page/ line number of change:

(Page 18; line 431-447) 

“Despite careful preparation, we acknowledged some limitations in our study that need to be identified. At first, the assumed period of 14 days might be a senseful approximation since the time between the onset of symptoms and testing is hardly detectable. Even though our study demonstrated the crucial role of centralized isolation in fighting the pandemic and may make other measures superfluous, but only if this measure is imposed at an early stage of virus spread, we also face a current challenge of identification all infections at an early stage of the outbreak. Detection of asymptomatic but infected people who had never been tested represented as an important matter that needed to be discussed. Besides, Asian countries with smaller territory sizes and accumulated gaining experiences of SAR-CoV-1 in the past had a greater benefit to tackle the spread of this infectious disease than European and American countries. Furthermore, Asian countries with “authoritarian government” and “collectivist culture” that pushed societal interests to the forefront might be easily adapted the lockdown policy. In contrast, European and American countries where “individualistic culture” is dominant are challenging to comply with strict enforcement of regulations. Finally, the heterogeneity in the analysis is unavoidable due to difficulties to reach a comparable definition of the time for each measure in countries such as the United States, Canada, and China. Also, there were differences in testing policies and the criteria of confirmes case in each country.”

Comment 6) Please include the following items when submitting your revised manuscript:

Author Response: Dear Editor, we have included all of these three required items in our submission. Thank you so much for your announcement.

Comment 7) When submitting your revision, we need you to address these additional requirements.

2. Please ensure links to all stated data sources have been included or cited in the methods section.

"The funders had no role in study design, data collection and analysis, decision to publish, or preparation of the manuscript,"

Author Response: Dear Editor, thank you so much for your announcement. We have addressed all of the above issues in our revised manuscript. 

Comment 8) At this time, please address the following queries:

a. Please clarify the sources of funding (financial or material support) for your study. List the grants or organizations that supported your study, including funding received from your institution.

d. If you did not receive any funding for this study, please state: “The authors received no specific funding for this work.”

Author Response: Dear Editor, thank you so much for your comment. We have addressed all of the above issues in our revised manuscript.

 

Reviewer #1: The article addresses a relevant and current topic. Explains the scientific background and rationale of the investigation. The methods section describes the setting, locations and relevant dates. The criteria of selection of countries were described. 

Comment 1) However, some elements could be better described. Data from the World Health Organization (March, 15) demonstrated that other countries have more cases or mortality, especially in Europe. Thus, it is important to explain why these countries were not selected.

Author Response: Dear Reviewer, we are thankful for your comment, and we are pleased to answer your question. From 70 days from January, 22nd 2020 (when the John Hopkins Coronavirus Resource Center began to collect the data of COVID-19) to the end of March 2020, we have collected data of countries with the highest number of daily confirmed cases. Among them, United States, Canada, and China, with the highest populations, were chosen due to their representative characteristic. Afterward, 5 Asian countries and 5 European countries with the percentage of days having the highest number of daily confirmed cases higher than 60% were chosen for later analysis. Additionally, Sweden, a North European country that decided to live in peace with the virus, was chosen as a European control country without both lockdown and centralized isolation. Details of the percentage of days having the highest number of daily confirmed cases of 13 countries were outlined in supplementary table 1.

Text Insertion (if applicable)/ page/ line number of change:

(Page 4; line 100-109) 

“In order to assess the impact of current control measures on the number of new cases, during 70 days from 22 January 2020 (when the John Hopkins Coronavirus Resource Center began to collect the data of COVID-19) to the end of March 2020, we have collected data of countries that had the highest number of daily confirmed cases. The United States, Canada, and China, with the highest population, were chosen due to their representative characteristic. Afterward, 5 Asian countries and 5 European countries with the percentage of days having the highest number of daily confirmed cases higher than 60% were chosen for later analysis. Additionally, Sweden, a North European country that decided to live in peace with the virus, was chosen as a European control country without both lockdown and centralized isolation. Details of the percentage of days having the highest number of daily confirmed cases of 13 countries were outlined in the S1 Table.”

(Page 21, line 517-518) 

S1 Table: 13 countries with the highest percentage of days having the highest number of daily confirmed cases

ID Country N (days) Percentage of days having the highest number of daily confirmed cases

1 China 70 100.00%

2 South Korea 70 100.00%

3 United State 70 100.00%

4 France 68 97.14%

5 Germany 65 92.86%

6 Italy 61 87.14%

7 United Kingdom 61 87.14%

8 Japan 60 85.71%

9 Canada 53 75.71%

10 Singapore 51 72.86%

11 Hong Kong 48 68.57%

12 Spain 47 67.14%

13 Taiwan 43 61.43%

Comment 2) Data relative to control measures could be better described. 

Author Response: Dear Reviewer, we are thankful for your comments. An additional paragraph and supplementary table 5 (attached excel file) had been added to outline the detail of each control measures as below: 

Text Insertion (if applicable)/ page/ line number of change:

(Page 5-6; line 135-159) 

“In China, Taiwan, and Hong Kong, all suspected and confirmed cases were isolated in the healthcare facilities and could be ruled out after a negative of SARS-CoV-2 tests with at least 24-hour intervals regardless of the days of isolation. In South Korea and Singapore, a mandatory of 14 days of isolation was applied for all suspected and confirmed cases of COVID-19. In Japan, only suspected and confirmed cases with related symptoms were isolated at the hospitals, whereas the remaining cases without any symptoms were strictly quarantined at medical camps, or hotels under medical staff supervision and during a required time. Even though the policy of centralized isolation of all confirmed cases was modified across these six countries, we assumed these control measures were comparable applied. Thus, centralized isolation of all confirmed cases is defined as a quarantine of all confirmed cases in hospitals, medical camps, or hotels under the supervision of medical staff and during a required time. Closure of schools was applied with the suspension of all cultural, educational, sports, and teaching activities. Closure of public areas consisted of non-essential business closures such as stores, shopping centers, services, such as bars, restaurants, cinemas, theaters, and prohibition of mass gathering. Additionally, citizens were requested not to leave the house unless for essential activities, always keep a safe distance (for example, at least 6-feet) and work from home. Closure of cities involved the restrictions on travel with the suspension of all forms of transportation across cities. Closure of borders is defined as the official announcement of the governments regarding the prohibition of entry and exit of all individuals from all entries, including lands, seas, rails, and air routes. Particularly, to address the heterogeneity of the date of implementation of control measures in countries with various states and cities such as the United States, Canada, and China, we assumed that the first date of applying the lock-down strategies is the representative date for these countries. Details of the date of the control measures mentioned above in each city and state of the United States, Canada, and China with reliable references were summarized in S5 table.”

Comment 3) The documents used as a reference to identify control measures must be referenced in the text for readers to consult. It is necessary to revise these references. Data from Spain (table 1) for example point to the closure of schools on March 11. However, this only occurred in the entire country after March 14. The same situation applies to control relaxation measures. In Spain, the control reduction plan is published on April 28, 2020 with different phases in each autonomous community.

Author Response: Dear Reviewer, we are thankful for your kind detections. To address the heterogeneity of the date of implementation of control measures in countries with various states and cities, such as the United States, Canada, Spain, and China, we assumed that the first date of applying the lock-down strategies is the representative date for these countries. For example, with the closure of schools in Spain, we chose the first date that Spain used this measure in Madrid on 11th March 2020 (reference as below). As your suggestions, we have adjusted the data in table 1 and we have also added an Supplementary Table 7 with the link of reliable references for readers to consult. Additionally, we also re-analyzed our data, and the new findings are not modified. 

https://edition.cnn.com/world/live-news/coronavirus-outbreak-03-11-20-intl-hnk/h_00ffe686205aac4779223e09fb9bbd0e

Comment 4) Another aspect in relation to control measures is the difficulty of establishing a date and intensity of measures in continental countries such as China, Canada and the United States. I highlight the situation in the USA, where each state institutes different measures. This question should be clearer in the methods, results tables and limitations of the Study.

Author Response: Dear Reviewer, we are thankful for your comment and totally agree with this remark. To address the heterogeneity of the date of implementation of control measures in countries with various states and cities such as the United States, Canada, and China, we assumed that the first date of application of the lock-down strategies is the representative date for these countries. We have acknowledged this limitation and mentioned it in the limitation section of this manuscript. An additional paragraph had been added to clarify this matter.

Text Insertion (if applicable)/ page/ line number of change:

(Page 6; line 154-159) 

“Particularly, to address the heterogeneity of the date of implementation of control measures in countries with various states and cities, such as the United States, Canada, Spain, and China, we assumed that the first date of applying the lock-down strategies is the representative date for these countries. Details of the date of the control measures mentioned above in each city and state of the United States, Canada, Spain, and China with reliable references were summarized in S5 table.”

(Page 18 ; line 444-447) 

“Finally, the heterogeneity in the analysis is unavoidable due to difficulties to reach a comparable definition of the time for each measure in countries such as the United States, Canada, and China. Also, there were differences in testing policies and the criteria of confirmes case in each country.”

Comment 5) Another relevant aspect is that it would be important to describe what was considered centralized isolation of all confirmed cases in the method section. Guidelines recommend isolation for all cases in most countries (including the WHO guideline). Thus, it is important for reads understand how the researchers define the difference between countries and references documents (the isolations were compulsory?). Other factors to understand the control measures like testing strategies were not described, analysed in the method section or mentioned in the discussion section. This aspect is important because a country could define isolation of all cases and do not test their citizens adequately, for example, and do not isolate a great part of the infected citizens. The statistics analysis must be described better in a supplementary document or in the text. 

Author Response: Dear Reviewer, we totally agree with your remark regarding the modification of centralization of all confirmed cases according to each country. In China, Taiwan, and Hong Kong, all suspected and confirmed cases were isolated in the healthcare facilities and could be ruled out after a negative of SARS-CoV-2 tests with at least 24-hour intervals regardless of the days of isolation. In South Korea and Singapore, a mandatory of 14 days of isolation was applied for all suspected and confirmed cases of COVID-19. In Japan, only suspected and confirmed cases with related symptoms were isolated at the hospitals, whereas the remaining cases without any symptoms were strictly quarantined at medical camps, or hotels under medical staff supervision and during a required time. Even though the policy of centralized isolation of all confirmed cases was modified across these six countries, we assumed these control measures were comparable applied. Thus, centralized isolation of all confirmed cases is defined as a quarantine of all confirmed cases in hospitals, medical camps, or hotels under the supervision of medical staff and during a required time. An additional paragraph had been added to clarify this issue as below:

Text Insertion (if applicable)/ page/ line number of change:

(Page 5; line 135-146) 

In China, Taiwan, and Hong Kong, all suspected and confirmed cases were isolated in the healthcare facilities and could be ruled out after a negative of SARS-CoV-2 tests with at least 24-hour intervals regardless of the days of isolation. In South Korea and Singapore, a mandatory of 14 days of isolation was applied for all suspected and confirmed cases of COVID-19. In Japan, only suspected and confirmed cases with related symptoms were isolated at the hospitals, whereas the remaining cases without any symptoms were strictly quarantined at medical camps, or hotels under medical staff supervision and during a required time. Even though the policy of centralized isolation of all confirmed cases was modified across these six countries, we assumed these control measures were comparable applied. Thus, centralized isolation of all confirmed cases is defined as a quarantine of all confirmed cases in hospitals, medical camps, or hotels under the supervision of medical staff and during a required time.

Comment 6) The results were well described, however it could be necessary to correct some results (changes in the definition of date of implementation of the national intervention and analysis of other countries).

Author Response: Dear Reviewer, we are thankful for your comment and totally agree with this remark. After modification of dates of implementation of the national interventions, updated analysis results and updated figures had been added to address these modifications. 

Text Insertion (if applicable)/ page/ line number of change:

(Page 9; line 238-239) 

Table 1: Date of implementation of the national intervention at the moment of the 10th confirmed cases

(Page 11; line 263) 

Table 2. Decreasing new case indices of countries by intervention solution

(Page 13; line 286-287) 

Figure 1: Analysis of anti-epidemic solutions according to the number of COVID-19 cases trajectories.

Comment 7) The discussion section could explore other hypothesis and aspects that could justify differences. All countries in group A were European or Americans with occidental culture. The group B has only Asiatic regions/countries. The experience with SAR-COV 1 was not mentioned like one aspect of difference between Asiatic and European and American Countries results. Different territory sizes and governance differences were not mentioned too (Asiatic regions were smaller and some are islands - these are importants factors in epidemic control). It could be important to describe the differences.

Author Response: Dear Reviewer, we totally agree with your comprehensive comments. Based on your suggestions, we made an update regarding these matters in the discussion section as below:

Text Insertion (if applicable)/ page/ line number of change:

(Page 18; line 438-444) 

Besides, Asian countries with smaller territory sizes and accumulated gaining experiences of SAR-CoV-1 in the past had a greater benefit to tackle the spread of this infectious disease than European and American countries. Furthermore, Asian countries with “authoritarian government” and “collectivist culture” that pushed societal interests to the forefront might be easily adapted the lockdown policy. In contrast, European and American countries where “individualistic culture” is dominant are challenging to comply with strict enforcement of regulations.

Comment 8) The study limitations could be better described (difficulties to defines the time of each measure, cases criteria definitions in which country, testing policies, governance differences etc). The conclusions were well described and were adequate to the results.

Author Response: Dear Reviewer, we are thankful for your remark. Based on your corrections, we have updated the limitation sections as below:

Text Insertion (if applicable)/ page/ line number of change:

(Page 18; line 431-447) 

“Despite careful preparation, we acknowledged some limitations in our study that need to be identified. At first, the assumed period of 14 days might be a senseful approximation since the time between the onset of symptoms and testing is hardly detectable. Even though our study demonstrated the crucial role of centralized isolation in fighting the pandemic and may make other measures superfluous, but only if this measure is imposed at an early stage of virus spread, we also face a current challenge of identification all infections at an early stage of the outbreak. Detection of asymptomatic but infected people who had never been tested represented as an important matter that needed to be discussed. Besides, Asian countries with smaller territory sizes and accumulated gaining experiences of SAR-CoV-1 in the past had a greater benefit to tackle the spread of this infectious disease than European and American countries. Furthermore, Asian countries with “authoritarian government” and “collectivist culture” that pushed societal interests to the forefront might be easily adapted the lockdown policy. In contrast, European and American countries where “individualistic culture” is dominant are challenging to comply with strict enforcement of regulations. Finally, the heterogeneity in the analysis is unavoidable due to difficulties to reach a comparable definition of the time for each measure in countries such as the United States, Canada, and China. Also, there were differences in testing policies and the criteria of confirmes case in each country.”

 

Reviewer #2: The authors attempt to assess the effectiveness of nonpharmaceutical interventions in fighting the SARS-CoV-2 pandemic. The study incorporates 13 countries and compares their strategies. From the methodological point of view, infections are analyzed over time, whilst testing the influence of different measures on the temporal development of infections. The authors conclude that centralized isolation of confirmed infected individuals is an effective strategy (on condition that is implemented early) rather than “lockdowns”. The study is very interesting and of high quality from a methodological perspective. The simultaneous comparison of different countries (with different economic and cultural background) is to be welcomed. However, there are some issues I would recommend to adress in a revision (“major revisions”), especially with respect to the selection of countries and the interpretation of the results (see below).

MAJOR COMMENTS

Comment 1) Again, I have to express my happiness about the comparison of different countries with different containment strategies. However, there is an important issue regarding the selection of these countries: With respect to the “lockdown” policy, the selected countries in the study can be divided into three groups (not to be confused with the classification made by the authors on page 8): A) Asian countries WITH strict “lockdowns”, B) Asian countries WITHOUT strict “lockdowns” and C) European and North American countries WITH strict “lockdowns”. However, there is one North European country (Sweden) which did NOT impose a “lockdown” as well (Note: We have to keep in mind that there have been several measures against SARS-CoV-2 spread both mandatory and voluntary – it is a common “fairy tale” that Sweden did nothing against the virus! Furthermore, assessing the Sweden strategy is complicated as, in the first wave of infections, Sweden had a much higher COVID mortality than its neighbours but a lower mortality compared to France, Italy, or Belgium with hard “lockdown” measures). But I would strongly recommend including Sweden into the analysis as a control group (European country WITHOUT “lockdown” AND without centralized isolation).

See this paper on Sweden by Born et al. 2020: `.

Author Response: Dear Reviewer, we are thankful for your valuable suggestion, and we totally agree with this remark. Using Sweden as a control country have helped to reinforce and demonstrate the importance of the implementation of the five control measures such as (a) centralized isolation of all confirmed cases, (b) closure of schools, (c) closure of public areas, (d) closure of cities, and (e) closure of borders. Our analysis showed that when other countries began to start the preventive methods, Sweden, without any implementation of control measures, continued to increase the number of new cases. While Spain, Italy, the United Kingdom, Germany, and France lessened the day of achieving the highest new cases (day 29th, day 30th, day 47th, day 55th, and day 65th, respectively), Sweden reached the peak of cases in day 118th with 62,728 confirmed cases according to government reports and 560,895 estimates. Additional analysis, writing, and update of figure 1 were added according to the recommendations and comments of the Reviewer as below:

Text Insertion (if applicable)/ page/ line number of change:

(Page 4; line 91-94) 

“Thus, our study made an effort to summarize and highlight distinctive features of significant control measures among 14 particular countries, including China, Hong Kong, Taiwan, Singapore, Korea, Japan, the United States, France, Germany, the United Kingdom, Canada, Italy, Spain, and Sweden.”

(Page 4; line 106-108) 

“Additionally, Sweden, a North European country that decided to live in peace with the virus, was chosen as a European control country without both lockdown and centralized isolation.”

(Page 4-5; line 115-124) 

“Regarding Sweden, since the testing policy primarily focused on citizens with symptoms associating with COVID-19 infection and requirement of the inpatient hospital care and (or) elderly care, whereas people with mild symptoms were skipped for contacting the healthcare[3], the number of new confirmed cases of Sweden reported from JHU might be lower than the actual new confirmed cases in reality. To address this limitation, we have estimated the number of new confirmed cases in Sweden by using the number of new confirmed cases of neighboured European countries such as Spain, Italy, the United Kingdom, France, and Germany at the moment without application of any control measures. Our analysis revealed that the actual new confirmed cases of Sweden, in reality, was 8.47 times higher than the report of JHU (S2 Table).”

(Page 8; line 234-235) 

“Sweden was excluded from all of the above control measures since this country denied to against the virus (Table 1).”

(Page 13; line 265-270) 

“Based on the implementation of methodsa,b,c,d,e, this study categorized 14 countries into two groups, group A and group B (Figure 1). Group A included Asian countries with strict lockdown policies (Taiwan, Hong Kong, Singapore, Japan, and China) and Asian countries without strict lockdowns (South Korea); they applied methoda. Group B consisted of European and North American countries with strict lockdown policies (Canada, France, Germany, Spain, Italy, United Kingdom, and the United States) and Sweden; they did not apply methoda”

(Page 13, line 286-287) 

Figure 1: Analysis of anti-epidemic solutions according to the number of COVID-19 case trajectories.

(Page 14; line 304-310) 

As for Sweden, the number of estimated total confirmed cases (dash line) was quite comparable to that of countries in group B when the number of new cases has not reached its peak. In which Spain, Italy, the United Kingdom, Germany, and France lessened the day of achieving the highest new cases (day 29th, day 30th, day 47th, day 55th, and day 65th, respectively), Sweden reached the peak of cases in day 118th with 62,728 confirmed cases according to government reports and 560,895 estimates. The results show that when Sweden did not apply any measuresa,b,c,d,e and the peak time was 1.8 – 4.1 times longer than other countries.

(Page 20) 

S2 Table: The average value of the ratio of daily new case between the 5 countries and Sweden.

Comment 2) As the analysis incorporates infection/fatalities data for 13 countries, it is reasonable to use a data source which provides all this case data for each country. Like in many studies towards the effectiveness of nonpharmaceutical interventions, the database from the Johns Hopkins University (JHU) was used. This data does, of course, not contain information about the true infection date (which is unknown in the majority of cases). By consequence, valid analyses of the effectiveness of nonpharmaceutical interventions must incorporate an appropriate delay between infection an the (possible) impact of measures. In the present study, the authors consider this problem by referring to the 95% percentile of the SARS-CoV-2/COVID-19 incubation period from the literature. The assumed period of 13 days might be a senseful approximation. However, this delay consists of (at least) three time periods: A) time between infection and onset of symptoms (if any; incubation period), B) time between onset of symptoms (if any) and testing, C) time between testing and reporting in official statistics (including JHU database). Note that these time periods may overlap. The authors should clarify this and should comment how this assumption on the delay might affect the results.

For further information on the issue of reporting delays etc., see for example the following literature:

Flaxman et al 2020 (https://doi.org/10.1038/s41586-020-2405-7) and Homburg 2020 (https://doi.org/10.1515/ev-2020-0010) also use JHU data for international comparisons of NPI effectiveness, but they trace infection dates back from reported death cases. A similar method is used by Wood 2020 (https://arxiv.org/pdf/2005.02090.pdf). A critique on former NPI studies using the JHU data (explicitly discussing the “reporting delay problem”) can be found in Wieland 2020 (https://doi.org/10.1016/j.ssci.2020.104924).

Author Response: Dear Reviewer, we are thankful for your kind comment and completely agree with the declaration that the assumed period of 13 days might be a senseful approximation, and this delay consists of (at least) three time periods: A) time between infection and onset of symptoms (if any; incubation period), B) time between the onset of symptoms (if any) and testing, C) time between testing and reporting in official statistics. To solve this issue, we conducted the study with the hypothesis that the time between testing and reporting in official statistics could be fluctuating from one day to four days with the details were described in supplementary table 6. After the analysis, the results showed that the data changes due to the fluctuation of time between testing and reporting in official statistics led to insignificant differences of 0-3.3% (CI 95%). Thus, we decided to select day 13th as a time point for analysis. An additional paragraph in the method section and in the limitation section of the discussion as well as supplementary table 6 have been added to clarify this matter.

Text Insertion (if applicable)/ page/ line number of change:

(Page 7; line 194-207) 

“From the literature, the average incubation period for SARS-Cov-2 is 5.2 days with the 95% confidence interval of the distribution at 13 days8. However, we encountered a delay in the time reporting of data compared to the testing time in reality. There were some possible reasons for this. Firstly, there were 13 days of average incubation period as mentioned above. Secondly, the differences in time zones between 14 countries could lead to the differences in reporting the data to the public between JHU and these countries. Besides, from April 2020, the JHU database only updated data at the constant time from 3:30-4:00 pm (UTC). Thirdly, there was a lack of simultaneousness in reporting confirmed cases, recovered cases, and death cases of COVID-19. To solve this issue, we conducted the study with the hypothesis that the time between testing and reporting in official statistics could be fluctuating from one day to four days with the details were described in S6 table. After the analysis, the results showed that the data changes due to differences in time zones between JHU and 14 countries led to insignificant differences of 0-3.3% (CI 95%). Thus, we decided to select day 13th as a time point for analysis.“

(Page 18; line 431-433) 

“Despite careful preparation, we acknowledged some limitations in our study that need to be identified. At first, the assumed period of 14 days might be a senseful approximation since the time between the onset of symptoms and testing is hardly detectable.”

(Page 20) 

S6 Table: Insignificant data changes due to the fluctuation of time between testing and reporting in official statistics

Comment 3) There is an issue with respect to the interpretation of the effectiveness of CENTRALIZED isolation: “[…] is not sufficient when we let the confirmed cases be self-isolated at home. The rationale behind this is that these individuals continue to do their daily activities such as buying food, entering local pharmacies, and patronizing other businesses, which might result in an increase in the risk of community spread.” (page 11, lines 323-326). I cannot assess the practical implementation of isolation in Singapore or Taiwan but, in European countries, domestic quarantine means a strict stay-at-home order which is controlled by health authorities. Is the centralized isolation strategy efficient because it is CENTRALIZED or because other types of isolation (such as domestic quarantine) are not or cannot be controlled? This is a more general issue, as, in the “Discussion” section, measures are supported because of their rigour: “However, as human nature dictates, adherence to isolation and other strategies e.g social distancing and wearing masks, can be difficult when not strictly enforced. Thus, protocols for enhancing community compliance with the secondary measures to further contain the spread, must be studied” (page 13, lines 375-378). How does this conclusion match the results from several European countries with differing NPIs and differing levels of self-responsibility? See e.g point 1 with respect to Sweden. I would not agree that any measure should be on a volunteer basis, but, in democratic states under the rule of law, governments must count on the support of individuals. I would ask the authors to include these aspects in the discussion, because voluntary compliance to (reasonable) rules are an important foundation of democratic societies.

Author Response: Dear Reviewer, we totally agree with your remark regarding voluntary compliance to (reasonable) rules as a vital foundation of democratic societies. The effectiveness of centralized isolation of all confirmed cases in various countries in our study is based on if the governments supported their citizens to comply with the control measure. In our study, centralized isolation of all confirmed cases is defined as a quarantine of all confirmed cases in hospitals, medical camps, or hotels under medical staff supervision and during a required time. In China, Taiwan, and Hong Kong, all suspected and confirmed cases were isolated in the healthcare facilities and could be ruled out after a negative of SARS-CoV-2 tests with at least 24-hour intervals regardless of the days of isolation. In South Korea and Singapore, a mandatory of 14 days of isolation was applied for all suspected and confirmed cases of COVID-19. To further strengthen the measure's compliance, South Korea activated the criminal penalty on violating the isolation order on February 26th, 20201. Besides isolating all suspected cases and confirmed cases at designated locations, Hong Kong also uniquely monitored people who quarantined at their homes by electrical wristbands2. In contrast, the USA, Canada, the UK, France, Germany, Italy, and Spain did not use centralized isolation of all confirmed cases. Furthermore, in European and American countries where “individualistic culture” is dominant, this is difficult to comply with strict enforcement of regulations. In contrast, Asian countries with “authoritarian government” and “collectivist culture” that pushed societal interests to the forefront might be easily adapted the lockdown policy.

1http://www.korea.kr/common/download.do?fileId=190536078&tblKey=GMN

2https://www.news.gov.hk/eng/2020/02/20200203/20200203_150504_450.html?type=category&name=covid19&tl=t

Text Insertion (if applicable)/ page/ line number of change:

(Page 5; line 135-144) 

“In China, Taiwan, and Hong Kong, all suspected and confirmed cases were isolated in the healthcare facilities and could be ruled out after a negative of SARS-CoV-2 tests with at least 24-hour intervals regardless of the days of isolation. In South Korea and Singapore, a mandatory of 14 days of isolation was applied for all suspected and confirmed cases of COVID-19. In Japan, only suspected and confirmed cases with related symptoms were isolated at the hospitals, whereas the remaining cases without any symptoms were strictly quarantined were strictly quarantined at medical camps, or hotels under medical staff supervision and during a required time. Even though the policy of centralized isolation of all confirmed cases was modified across these six countries, we assumed these control measures were comparable applied.” 

(Page 17; line 440-444) 

“Furthermore, Asian countries with “authoritarian government” and “collectivist culture” that pushed societal interests to the forefront might be easily adapted the lockdown policy. In contrast, European and American countries where “individualistic culture” is dominant are challenging to comply with strict enforcement of regulations.”

MINOR COMMENTS

Comment 4) The criteria for county selection appear reasonable in most cases, but please describe (just 1-2 sentences) what is meant by “having a close relationship with the outbreak's origin (Wuhan, China)”.

Author Response: Dear Reviewer, after revising the manuscript, we decide to remove this sentence and insert a new paragraph to clarify the selection strategy of the targeted countries in our study as below:

Text Insertion (if applicable)/ page/ line number of change:

(Page 4; line 100-109) 

“In order to assess the impact of current control measures on the number of new cases, during 70 days from 22 January 2020 (when John Hopkins Coronavirus Resource Center began to collect the data of COVID-19) to the end of March 2020, we have collected data of countries that had the highest number of daily confirmed cases. The United States, Canada, and China, with the highest population, were chosen due to their representative characteristic. Afterward, 5 Asian countries and 5 European countries with the percentage of days having the highest number of daily confirmed cases higher than 60% were chosen for later analysis. Additionally, Sweden, a North European country that decided to live in peace with the virus, was chosen as a European control country without both lockdown and centralized isolation. Details of the percentage of days having the highest number of daily confirmed cases of 13 countries were outlined in the S1 Table.”

(Page 21; line 517-518) 

S1 Table: 13 countries with the highest percentage of days having the highest number of daily confirmed cases

Comment 5) Centralized isolation is identified as an effective strategy in fighting the pandemic and may make other measures superfluous, but only if this measure is imposed at an early stage of virus spread (“On the other hand, the application of centralized isolation would no longer be effective without other solutions such as closing public places, schools, cities, or borders; when the number of active cases was over 400”; page 9, lines 265-267). We know that SARS-CoV-2 cases are underestimated everywhere because of an unknown “dark figure” of asymptomatic but infected people who have never been tested. However, official reported cases are the only data source available, and, according the WHO, “asymptomatically infected individuals are much less likely to transmit the virus than those who develop symptoms” (http://www.emro.who.int/health-topics/corona-virus/transmission-of-covid-19-by-asymptomatic-cases.html). Is it realistic to identify all infections at an early stage of the pandemic? I would recommend that the authors should reflect this issue, just in 1-2 sentences.

Author Response: Dear Reviewer, your above comment was exact and comprehensive. It’s obvious that identify all infections at an early stage of the pandemic is challengeable and even irrealistic. We have addressed these ideas in the limitation section as below:

Text Insertion (if applicable)/ page/ line number of change:

(Page 18; line 433-438)

Even though our study demonstrated the crucial role of centralized isolation in fighting the pandemic and may make other measures superfluous, but only if this measure is imposed at an early stage of virus spread, we also face a current challenge of identification all infections at an early stage of the outbreak. Detection of asymptomatic but infected people who had never been tested represented as an important matter that needed to be discussed.

Comment 6) Page 6, line 156: “Since vaccines and target drugs are not available…” should be more like “Since vaccines and target drugs were not available in March 2020…”. A similar issue is on page 11, line 319: “…while vaccination and approved drugs for COVID-19 are still unavailable…”. Note that I received the review invitation on December 12, 2020, and I don’t know when the manuscript was written and submitted. Now, there is a vaccine.

Author Response: Dear Reviewer, we totally agree to your comments. An adjustment of these two sentences had been inserted as below:

Text Insertion (if applicable)/ page/ line number of change:

(Page 6; line 171-172) 

“Since vaccines and target drugs were not available in March 2020, current interventions can only control the number of new cases.”

(Page 16; line 368-370)

“Currently, while vaccination and approved drugs for COVID-19 require time to assess its efficaciousness, the management of the sources of infection, as well as routes of transmission, remain the vital factors to control the pandemic.”

Comment 7) The authors should refer some previous literature towards analyses of the effectiveness of NPIs, especially country comparisons. See the literature in point 2 and, e.g., Chaudry et al 2020 (https://doi.org/10.1016/j.eclinm.2020.100464).

Author Response: Dear Reviewer, thank you so much for your kind recommendation. After reading this paper, we have added some sentences to refer to the previous findings towards the effectiveness of NPIs as below:

Text Insertion (if applicable)/ page/ line number of change:

(Page 17; line 395-399) 

“In the same manner, our findings were in line with the report of Chaudhry et al., which identified that the days to partial or full lockdown and the day to any border closures were the two significant predictors associating with the total number of reported cases per million14. Interestingly, this author also figured out that a full lockdown policy was a supporting factor in increasing the number of recovered cases.”

Comment 8) On page 13, lines 385-386, the authors write that “lockdown” policies have “a negative impact on the society and economy”. Like the majority of scientists, I would agree with that. However, it may be a good idea to cite some literature on negative impacts of “lockdowns” already in the “Introduction” section.

For example, Miles et al. 2020 (https://doi.org/10.1111/ijcp.13674) balance the costs and benefits of “lockdowns” and find that the economic impacts are more harmful than their benefits (economic consequences). Adams-Prassl et al 2020 (https://hceconomics.uchicago.edu/research/working-paper/impact-coronavirus-lockdown-mental-health-evidence-us) identify severe impacts of stay-at-home orders on mental health (psychosocial consequences).

Author Response: Dear Reviewer, we are thankful for your kind comments with the two valuable suggested articles. Additional sentences have been inserted into our manuscript to mention the negative impacts of lockdown policy in the introduction section as below:

Text Insertion (if applicable)/ page/ line number of change:

(Page 3-4; line 83-91) 

“However, these lockdown policies are hurting with negative impacts in multiple aspects. Miles et al. determined that the lockdown cost was 40% higher than the highest benefits from avoiding the worst mortality case scenario at full life expectancy tariff1. For mental health, Adams-Prassl et al. recognized that citizens living in states with implementation of lockdown scored 0.085 standard deviations lower on the standardized WHO-5 mental health index compared to those living in states without lockdown strategy2. Despite its negative impacts, potentially measuring implementation also effectively proved its ability to control this pandemic and remains a question that we hope to elucidate.”

Comment 9) Of course, it’s a matter of style but maybe the authors should use the term “nonpharmaceutical interventions” at least once in the text and/or the abstract (because this is the “de facto official” term for containment measures in the international literature). But, keep in mind that this is a “style question” :)

Author Response: Dear Reviewer, we agree with your suggestion regarding the use of the term “non-pharmaceutical interventions.” Adjustments had been inserted in the manuscript as below: 

Text Insertion (if applicable)/ page/ line number of change:

(Page 2; line 48-50) 

“However, when the number of cases increased with the apparition of new clusters, coordination between centralized isolation and non-pharmaceutical interventions would facilitate controlling the crisis efficiently.”

(Page 3; line 80-81) 

“A list of non-pharmaceutical interventions at different levels and various strategies had been established to prevent and mitigate the continual spread of COVID-19…”

(Page 17; line 389) 

“Our result determined that all non-pharmaceutical interventions, including the closure of schoolsb..”

(Page 19; line 454-457) 

“However, when the number of cases increased with the apparition of new clusters, coordination between centralized isolation and non-pharmaceutical interventions would facilitate controlling the crisis efficiently.”

Comment 10) The figures are a little bit blurred in the review version, but this may be due to graphics compression during the processing of the manuscript?!

Author Response: Dear Reviewer, thank you so much for this remark. As you said, it might be due to the graphics compression during the processing of the manuscript. However, we have adjusted the resolution of our figures to achieve a better visualization. Please recheck it and I hope it responses to your requirement.

---

## [Decision Letter · Decision Letter 1]

31 Mar 2021

PONE-D-20-33060R1

Early centralized isolation strategy for all confirmed cases of COVID-19 remains a core intervention to disrupt the pandemic spreading significantly

PLOS ONE

Dear Dr. Huy,

Thank you for submitting your manuscript to PLOS ONE. After careful consideration, we feel that it has merit but does not fully meet PLOS ONE’s publication criteria as it currently stands. Therefore, we invite you to submit a revised version of the manuscript that addresses the points raised during the review process.

In particular, please consider the ways that Reviewer #3 has suggested to improve the manuscript.

We look forward to receiving your revised manuscript.

Kind regards,

Siew Ann Cheong, Ph.D.

Academic Editor

PLOS ONE

Journal Requirements:

Reviewers' comments:

Reviewer's Responses to Questions

**Comments to the Author**

1. If the authors have adequately addressed your comments raised in a previous round of review and you feel that this manuscript is now acceptable for publication, you may indicate that here to bypass the “Comments to the Author” section, enter your conflict of interest statement in the “Confidential to Editor” section, and submit your "Accept" recommendation.

Reviewer #1: All comments have been addressed

Reviewer #3: (No Response)

2. Is the manuscript technically sound, and do the data support the conclusions?

Reviewer #1: Yes

Reviewer #3: Partly

3. Has the statistical analysis been performed appropriately and rigorously? 

Reviewer #1: I Don't Know

Reviewer #3: Yes

4. Have the authors made all data underlying the findings in their manuscript fully available?

Reviewer #1: Yes

Reviewer #3: Yes

5. Is the manuscript presented in an intelligible fashion and written in standard English?

Reviewer #1: Yes

Reviewer #3: Yes

6. Review Comments to the Author

Reviewer #1: All suggestions and recommendations were accepted by the authors and adjustments were made. I am not a native English speaker and the analysis of writing in "standard English" may be inadequate. Statistical analysis seems adequate, but I am not an expert. So, I prefer an answer that I'm not sure.

Reviewer #3: Please see the attached review report.

7. PLOS authors have the option to publish the peer review history of their article (what does this mean?). If published, this will include your full peer review and any attached files.

Reviewer #1: **Yes: **Alexandre Medeiros de Figueiredo

Reviewer #3: No

---

## [Author Response · Author response to Decision Letter 1]

23 Apr 2021

April 23th, 2021

PLOS ONE PONE-D-20-33060 R1

Early centralized isolation strategy for all confirmed cases of COVID-19 remains a core intervention to disrupt the pandemic spreading significantly

Dear Editors and Reviewers,

We are most grateful to the Editor, the Associate Editor, and the Reviewer for providing helpful and constructive comments regarding our manuscript. We have taken all of these comments into account and hereby submit a revised manuscript with the changes underlined.

We responded to all of the comments as indicated below, and we hope that our explanations and revisions will be deemed satisfactory.

Sincerely,

Associate Nguyen Tien Huy, M.D, Ph.D.

School of Tropical Medicine and Global Health, Nagasaki University, Nagasaki 852-8523, Japan 

E-mail: tienhuy@nagasaki-u.ac.jp

 

Reviewer #3: Summary 

This paper aims to show that the early centralized isolation is the major factor out of five control measures in disrupting the pandemic’s spread. In order to assess the impact, a number of metrics has been used: new case indices, absolute effect average, absolute effect cumulative and relative effect average, posterior probability of a casual effect. 

Overall remarks 

The paper is well-written and well-organized. The main objective is clearly defined. However, the way of achieving the objective is not sufficient. In particular, the empirical analysis is not enough to reach the conclusion. Authors should incorporate some important metrics which are widely used to measure the spread of infectious disease. I believe that the authors can improve the paper by taking into account the suggestions provided below. 

Comment 1) We know that reproduction rate is statistically very important metric that measures the ability of any disease to spread. It would be great if authors can analyze time-varying reproduction number (R(t)) in pre-intervention and post-intervention periods. 

Author Response: Dear Reviewer, thank you so much for your kind comments. Assessment of time-varying reproduction number (R(t)) in pre-intervention and post-intervention periods is undeniable an excellent approach to verify and emphasize our findings. Based on your suggestion, we have performed the additional analysis. An update paragraph and table concerning the aformentioned analysis had been inserted as below:

Text Insertion (if applicable)/ page/ line number of change:

(Page 16; line 361-371) 

“..Regarding reproduction rate, it was calculated by using the number of death cases. However, during the early stage of the pandemic, the number of death cases was low, which consequently led to difficulties in estimating the reproduction rate. Therefore, we estimated the missing values by estimating the correlation between the reproduction rate and the ratio of new cases (dayn) and new cases (day n-1). Table 3 showed that on the 13th day, countries in group B, which did not apply any control measures, had an average reproduction rate of 2.23 (0.38). However, after applying measures such as the closing of schools, public areas, and borders, this index achieved a reduction from 1.2 to 2.5 points. For countries in group A, the average reproduction rate was 1.28 (0.64) before applying closing public areas, cities, or borders. However, when they used the above measures, this rate fluctuated between 0.093 and 1.1. In testing with the fluctuation from 1 to 4 days, the analysis showed similar statistical results with insignificant difference

(Page 17; line 372) 

Table 3: Absolute effect of the reproduction rate 

Country Absolute effect of the reproduction rate (95% CI)p

 13 days 12 days (– 1) 11 days (– 2) 10 days (– 3) 9 days (– 4)

SpainNo � bcde -1.9 (-2.3, -1.5)**** -1.9 (-2.3, -1.5)**** -1.9 (-2.3, -1.5)**** -1.9 (-2.3, -1.5)**** -1.9 (-2.3, -1.5)****

Spainbcde � bde -0.31 (-0.44, -0.19)**** -0.35 (-0.49, -0.21)**** -0.38 (-0.53, -0.23)**** -0.42 (-0.59, -0.25)**** -0.45 (-0.64, -0.27)****

Italy No � bcde -2.5 (-3.7, -1.4)**** -2.6 (-3.8, -1.5)**** -2.6 (-3.8, -1.5)**** -2.7 (-3.9, -1.5)**** -2.7 (-3.9, -1.6)****

United Kingdom No � bc -1.2 (-1.5, -0.99)**** -1.2 (-1.5, -1)**** -1.2 (-1.5, -1)**** -1.2 (-1.5, -1)**** -1.2 (-1.5, -0.99)****

United Kingdom bc � b -0.25 (-0.34, -0.16)**** -0.27 (-0.37, -0.17)**** -0.29 (-0.4, -0.18)**** -0.32 (-0.44, -0.2)**** -0.35 (-0.48, -0.21)****

Canada No � bce -1.3 (-1.4, -1.1)**** -1.3 (-1.4, -1.2)**** -1.3 (-1.4, -1.2)**** -1.3 (-1.4, -1.2)**** -1.3 (-1.4, -1.2)****

United States No � bcde -1.7 (-1.9, -1.4)**** -1.7 (-1.9, -1.4)**** -1.7 (-1.9, -1.4)**** -1.7 (-1.9, -1.4)**** -1.7 (-1.9, -1.4)****

France No � bce -1.7 (-2, -1.4)**** -1.7 (-2, -1.4)**** -1.7 (-2, -1.4)**** -1.7 (-2, -1.4)**** -1.7 (-2, -1.4)****

GermanyNo � bce -1.6 (-1.9, -1.4)**** -1.6 (-1.8, -1.4)**** -1.6 (-1.8, -1.4)**** -1.6-1.3 (-1.8, -1.4)**** -1.6 (-1.8, -1.4)****

Chinabcd � abcd -2 (-2.6, -1.5)**** -2 (-2.5, -1.6)**** -2.1 (-2.5, -1.6)**** -2.1 (-2.6, -1.7)**** -2.1 (-2.6, -1.7)****

Chinaabcd � abd 0.5 (0.43, 0.56)**** 0.52 (0.41, 0.63)**** 0.52 (0.41, 0.6)**** 0.5 (0.38, 0.62)**** 0.47 (0.32, 0.61)****

Chinaabd � abe -0.4 (-0.52, -0.29)**** -0.5 (-0.55, -0.45)**** -0.49 (-0.54, -0.44)**** -0.47 (-0.52, -0.42)**** -0.43 (-0.48, -0.38)****

KoreaNo �abc -1.1 (-1.7, -0.6)**** -1.2 (-1.7, -0.64)**** -1.2 (-1.8, -0.68)**** -1.3 (-1.8, -0.74)**** -1.3 (-1.8, -0.79)****

Japan No � a -0.093 (-0.23, 0.036)0.082 -0.11 (-0.24, 0.02)* -0.13 (-0.26, 0.004)* -0.14 (-0.27, -0.013)* -0.15 (-0.28, -0.025)**

Japana � abe 0.52 (0.4, 0.64)**** 0.41 (0.3, 0.52)**** 0.41 (0.29, 0.52)**** 0.4 (0.29, 0.51)**** 0.38 (0.27, 0.5)****

Japanabe � ae -0.68 (-0.8, -0.56)**** -0.78 (-0.94, -0.61)**** -0.73 (-0.9, -0.57)**** -0.68 (-0.86, -0.5)**** -0.62 (-0.81, -0.42)****

Japanae � acde -0.57 (-0.82, -0.33)**** -0.35 (-0.48, -0.24)**** -0.38 (-0.51, -0.26)**** -0.42 (-0.56, -0.28)**** -0.46 (-0.62, -0.32)****

Singaporea � abce -0.43 (-0.61, -0.24)**** -0.39 (-0.57, -0.21)*** -0.35 (-0.52, -0.17)**** -0.3 (-0.48, -0.13)**** -0.26 (-0.42, -0.086)**

Hong Konga � abce -0.49 (-0.67, -0.32)**** -0.51 (-0.68, -0.35)**** -0.53 (-0.69, -0.36)**** -0.54 (-0.7, -0.39)**** -0.56 (-0.71, -0.4)****

Taiwanab � a -0.38 (-0.43, -0.33)**** -0.36 (-0.41, -0.31)**** -0.33 (-0.39, -0.28)**** -0.31 (-0.37, -0.26)**** -0.29 (-0.34, -0.24)****

Taiwana �ace -0.31 (-0.38, -0.25)**** -0.39 (-0.53, -0.26)**** -0.31 (-0.37, -0.26)**** -0.32 (-0.38, -0.27)**** -0.41 (-0.55, -0.28)****

* < 0.05 ** < 0.01 *** < 0.001 **** < 0.0001

Comment 2) One of the key indicators of any pandemic spread is the doubling time. It also measures the possibility of achieving flattened epidemic curve. Thus, it would be helpful if authors can present some result based on doubling time. 

I believe that inclusion of these evidences will not only make the paper statistically stronger but also more conclusive. 

Author Response: Dear Reviewer, we are thankful for your kind suggestion. Based on your valuable recommendation, we have analyzed the doubling time. An additional paragraph together with an illustrative figure had been added as below:

Text Insertion (if applicable)/ page/ line number of change:

(Page 16; line 344-357) 

“..Additionally, in order to verify and emphasize our above findings, the two important metrics, such as doubling time, as well as time-varying reproduction number in pre-intervention and post-intervention periods, which are widely used to evaluate the spread of infectious disease, were also calculated. For doubling time, Figure 3 demonstrated that the United States suffered the highest doubling time, which was 14 times. In contrast, Taiwan, with its reasonably preventive policies, had the lowest doubling time in our analysis. Indeed, 100 days since 10th confirmed cases, group A, which was using the centralized isolation of all confirmed cases, resulted in a lower doubling time: Japan and Singapore’s doubling times were 9 and 10, respectively; and the other four countries in this group had the doubling times under 8. In reverse, the doubling time in group B seems to be more serious, with the excess of the 8th doubling time just after 20 - 40 days. Notably, the 11th doubling time presented from day 30th to day 70th since the 10th case. In China and South Korea, without centralized isolation of all confirmed cases, their 7th doubling times were just after 20th day and 34th day, respectively, in the pandemic's early stage. However, after applying the aforementioned measure, their doubling times just increased for one more scale after 70 days.. “

(Page 16; line 359) 

Figure 3: Doubling time analysis

Comment 3) This type of paper has tremendous scope of using various data visualization techniques to display the results. For example, authors should include more figures, charts, tables to display the changes related to the new case indices, absolute effect average, posterior probability and other indices. Similarly, wherever possible, try to use visualization tools instead of providing long elaboration. This will make the paper concise and visually appealing. 

Author Response: Dear Reviewers, we are thankful for this constructive comment. It obviously help to enhance the visualization of our current manuscript. As your suggestion, the two addtional figure which are related to the illustration of the new case indices and the absolute effect average had been added as below:

Text Insertion (if applicable)/ page/ line number of change:

(Page 10; line 253-254) 

“…Illustration of the change in new case index according to the application of different intervention strategies was represented in S1 Figure.”

(Page 10; line 256-257) 

S1 Figure: Change in new case index according to the application of different intervention strategies

(Page 10; line 266-268) 

“…Modification of absolute effect average in each country according to the application of different intervention strategies was depicted in S2 Figure.”

(Page 11; line 272-274) 

S2 Figure: Modification of absolute effect average in each country according to the application of different intervention strategies 

(Posterior probability of a causal effect; * < 0.05 , ** < 0.01, *** < 0.001, **** < 0.0001)

---

## [Decision Letter · Decision Letter 2]

25 May 2021

PONE-D-20-33060R2

Early centralized isolation strategy for all confirmed cases of COVID-19 remains a core intervention to disrupt the pandemic spreading significantly

PLOS ONE

Dear Dr. Huy,

Thank you for submitting your manuscript to PLOS ONE. After careful consideration, we feel that it has merit but does not fully meet PLOS ONE’s publication criteria as it currently stands. Therefore, we invite you to submit a revised version of the manuscript that addresses the points raised during the review process.

We look forward to receiving your revised manuscript.

Kind regards,

Siew Ann Cheong, Ph.D.

Academic Editor

PLOS ONE

Journal Requirements:

Reviewers' comments:

Reviewer's Responses to Questions

**Comments to the Author**

1. If the authors have adequately addressed your comments raised in a previous round of review and you feel that this manuscript is now acceptable for publication, you may indicate that here to bypass the “Comments to the Author” section, enter your conflict of interest statement in the “Confidential to Editor” section, and submit your "Accept" recommendation.

Reviewer #3: All comments have been addressed

2. Is the manuscript technically sound, and do the data support the conclusions?

Reviewer #3: Yes

3. Has the statistical analysis been performed appropriately and rigorously? 

Reviewer #3: Yes

4. Have the authors made all data underlying the findings in their manuscript fully available?

Reviewer #3: (No Response)

5. Is the manuscript presented in an intelligible fashion and written in standard English?

Reviewer #3: Yes

6. Review Comments to the Author

Reviewer #3: See the attached file

7. PLOS authors have the option to publish the peer review history of their article (what does this mean?). If published, this will include your full peer review and any attached files.

Reviewer #3: No

---

## [Author Response · Author response to Decision Letter 2]

14 Jun 2021

PLOS ONE PONE-D-20-33060 R2

Early centralized isolation strategy for all confirmed cases of COVID-19 remains a core intervention to disrupt the pandemic spreading significantly

Dear Editors and Reviewers,

We are most grateful to the Editor, the Associate Editor, and the Reviewer for providing helpful and constructive comments regarding our manuscript. We have taken all of these comments into account and hereby submit a revised manuscript with the changes underlined.

We responded to all of the comments as indicated below, and we hope that our explanations and revisions will be deemed satisfactory.

Sincerely,

Associate Nguyen Tien Huy, M.D, Ph.D.

School of Tropical Medicine and Global Health, Nagasaki University, Nagasaki 852-8523, Japan 

E-mail: tienhuy@nagasaki-u.ac.jp

 

Reviewer #3: Overall remarks

The revised version looks better now. The authors have included more charts/figures that make the paper visually more appealing. The analysis of doubling time and the reproduction rate have also been included. However, the work related to the reproduction rate (RR) lacks some important aspects. Authors need to improve that part. 

Comment 1) It is not clear which methodology was used for estimating the RR. The authors just mentioned that the RR was calculated by using the number of death cases. Exact name and relevant references are needed unless new method is proposed. 

Author Response: Dear Reviewer, thank you so much for your constructive comments. An additional paragraph has been added to clarify the methodology was used to estimate the RR, as well as the concept of RR and the relevant references

Text Insertion (if applicable)/ page/ line number of change:

(Page 16-17; line 368-374) 

“Regarding reproduction rate RR, it evaluates how effective the control measures are in mitigating the growth of the confirmed cases. This index is estimated by the average the population who become infected by one infectious person. Alternatively, it can be calculated through the number of mortality [12]. If RR is under 1, the spread of the virus is slowing.Whereas if the value is greater than 1, the virus spread is increasing [13]. Our study evaluated the effectiveness of intervention through RR before and after applying these solutions. The reproduction rate is obtained from the database of the “Our World In Data” (https://ourworldindata.org/coronavirus-data), a project output of the Oxford Martin Programme on Global Development from the University of Oxford.”

12. Dietz K. The estimation of the basic reproduction number for infectious diseases. Stat Methods Med Res. 1993;2: 23–41. doi:10.1177/096228029300200103

13. Toraih EA, Hussein MH, Elshazli RM, Fawzy MS, Houghton A, Tatum D, et al. Time-varying Reproductive Rates for SARS-CoV-2 and its Implications as a Means of Disease Surveillance on Lockdown Restrictions. Ann Surg. 2021;273: 28–33. doi:10.1097/SLA.0000000000004590

(Page 17; line 377-378) 

“Time-varying of reproduction rate for all the relevant countries was outlined in Figure 4.”

Comment 2) Before explaining empirical analysis, authors should explain the concept of RR and its importance (similarly for the doubling time analysis). Provide the formula of the corresponding metrics. 

Author Response: Dear Reviewer, we totally agree with your remark. Thank you so much for your kind comments. An additional paragraph has been added to clarify the methodology was used to analyze the doubling time, as well as the concept of doubling time and the relevant references

Text Insertion (if applicable)/ page/ line number of change:

(Page 16; line 348-354) 

“These two metrics are widely used to evaluate the spread of infectious diseases. The doubling time is the required time to duplicate the number of infected peoples. Given the exponential growth of an epidemic and the constant growth rate r, the doubling time needed is calculated as (ln 2)/r. Thus, an elevation in doubling time suggests a reduction in virus transmission. In our study, the doubling time was calculated by the days that the positive cases increase to double values or the days that the increase of positive cases reaches 100% [11].”

11. Painter PL, Marr AG. Inequality of Mean Interdivision Time and Doubling Time. Microbiology,. Microbiology Society,; 1967. pp. 155–159. Available: https://www.microbiologyresearch.org/content/journal/micro/10.1099/00221287-48-1-155

Comment 3) If possible, try to include figures of time-varying RR for all the relevant countries. 

Author Response: Dear Reviewer, thank you so much for your kind comments.We have imcluded the new Figure 4 into our submission to illustrate the time-varying RR for all the relevant countries.

Text Insertion (if applicable)/ page/ line number of change:

(Page 17; line 386) 

Figure 4: Absolute effect of Government policies on reproduction rate

---

## [Editor Report · Decision Letter 3]

18 Jun 2021

Early centralized isolation strategy for all confirmed cases of COVID-19 remains a core intervention to disrupt the pandemic spreading significantly

PONE-D-20-33060R3

Dear Dr. Huy,

We’re pleased to inform you that your manuscript has been judged scientifically suitable for publication and will be formally accepted for publication once it meets all outstanding technical requirements.

Kind regards,

Siew Ann Cheong, Ph.D.

Academic Editor

PLOS ONE
---

## [Editor Report · Acceptance letter]

24 Jun 2021

PONE-D-20-33060R3 

Early centralized isolation strategy for all confirmed cases of COVID-19 remains a core intervention to disrupt the pandemic spreading significantly 

Dear Dr. Huy:

I'm pleased to inform you that your manuscript has been deemed suitable for publication in PLOS ONE. Congratulations! Your manuscript is now with our production department. 

Kind regards, 

on behalf of

Dr. Siew Ann Cheong 

Academic Editor

PLOS ONE